# Fast Rank-1 Lattice Targeted Sampling for Black-box Optimization

**Yueming LYU**

Centre for Frontier AI Research (CFAR)
Institute of High Performance Computing (IHPC)
Agency for Science, Technology and Research (A*STAR)
1 Fusionopolis Way, #16-16 Connexis, Singapore 138632
Lyu_Yueming@cfar.a-star.edu.sg

## Abstract

Black-box optimization has gained great attention for its success in recent applications. However, scaling up to high-dimensional problems with good query efficiency remains challenging. This paper proposes a novel Rank-1 Lattice Targeted Sampling (RLTS) technique to address this issue. Our RLTS benefits from random rank-1 lattice Quasi-Monte Carlo, which enables us to perform fast local exact Gaussian processes (GP) training and inference with $O(n \log n)$ complexity w.r.t. $n$ batch samples. Furthermore, we developed a fast coordinate searching method with $O(n \log n)$ time complexity for fast targeted sampling. The fast computation enables us to plug our RLTS into the sampling phase of stochastic optimization methods. This improves the query efficiency while scaling up to higher dimensional problems than Bayesian optimization. Moreover, to construct rank-1 lattices efficiently, we proposed a closed-form construction. Extensive experiments on challenging benchmark test functions and black-box prompt fine-tuning for large language models demonstrate the query efficiency of our RLTS technique.

## 1 Introduction

Black-box optimization has gained great attention for its success in many recent applications, such as prompt fine-tuning for large language models [Sun et al., 2022b,a], policy search for robot control and reinforcement learning [Choromanski et al., 2019, Lizotte et al., 2007, Barsce et al., 2017, Salimans et al., 2017], automatic hyper-parameters tuning in machine learning problems [Snoek et al., 2012], black-box architecture search in engineering design [Wang and Shan, 2007], drug discovery [Negoescu et al., 2011] and accelerated simulation for scientific discovery [Maddox et al., 2021, Hernández-Lobato et al., 2017], etc. Many efforts have been made for black-box optimization in the literature, including Bayesian optimization (BO) methods [Srinivas et al., 2010, Gardner et al., 2017, Nayebi et al., 2019], stochastic optimization methods like evolution strategies (ES) [Back et al., 1991, Hansen, 2006, Wierstra et al., 2014b, Lyu and Tsang, 2021] and genetic algorithms [Srinivas and Patnaik, 1994, Mirjalili and Mirjalili, 2019].

Bayesian optimization usually builds a global (GP) model as a surrogate and provides queries by optimizing some acquisition functions [Snoek et al., 2012]. Although BO achieves good query efficiency for low-dimensional problems, it often fails to handle high-dimensional problems with large sample budgets [Eriksson et al., 2019]. The computation of GP with a large number of samples itself is expensive, and the internal optimization of the acquisition functions is challenging. Recently, Müller et al. [2021], Nguyen et al. [2022] builds a GP model for both the function value and the gradient and performs local Bayesian optimization. Although these methods improve the scalability of global BO, they usually cannot scale up to five hundred dimensional complex problems. This may

37th Conference on Neural Information Processing Systems (NeurIPS 2023).

be because the learned gradient heavily depends on the accuracy of the GP model. However, achieving an accurate GP model is challenging for high-dimensional problems. A slightly misspecified GP model may lead to a wrong estimated gradient due to the highly nonlinear acquisition functions.

On the other line, stochastic optimization methods, e.g., ES [Rechenberg and Eigen, 1973, Nesterov and Spokoiny, 2017], natural evolution strategies (NES) [Wierstra et al., 2014b], CMAES [Hansen, 2006], and implicit natural gradient optimizer (INGO) [Lyu and Tsang, 2021], typically sampling form Gaussian distribution and approximate the (natural) gradient for the update of the Gaussian distribution parameters for continuous optimization. These methods can scale up to higher dimensional problems compared with BO. However, the gradient approximation may have a large variance, especially for high-dimensional problems. Thus, the update direction may not be toward the descent direction, leading to inferior query efficiency.

To address high-dimensional black-box problems with good query efficiency, we propose a novel Rank-1 Lattice Targeted Sampling (RLTS) technique. Our RLTS has a $O(n \log n)$ time complexity, which is fast for plugging into the sampling phase of stochastic optimization methods. In this way, our methods can improve the query efficiency of stochastic optimization methods while addressing higher-dimensional problems than BO. Our contributions are summarized as follows:

- We propose a novel Rank-1 Lattice Targeted Sampling (RLTS) technique. Our RLTS builds a local GP with a random rank-1 lattice, which enables fast exact GP training and inference with $O(n \log n)$ time complexity w.r.t. $n$ batch samples. Furthermore, we develop a fast coordinate search that enables target sampling with $O(n \log n)$ time complexity.

- We propose a closed-form subgroup rank-1 lattice by considering the dual lattice regarding the integral approximation error of functions in Korobov space. Our rank-1 lattice has a more regular pattern of approximation error terms. Moreover, our subgroup rank-1 lattice capitalizes on constructing a circulant kernel Gram matrix benefit from its group property. This enables efficient $O(n \log n)$ computations in GP training/inference and fast candidate searching. In contrast, low-discrepancy QMC sequences, such as Sobol sequences or Halton sequences, lack these capabilities. In addition, our new closed-form rank-1 lattice may have potential applications in downstream tasks beyond black-box optimization.

- We plug our RLTS into the sampling phase at each step of stochastic optimization methods to improve query efficiency. In this way, during the optimization procedure, our RLTS sampling from an updated promising region instead of a fixed one at each step. This approach can scale up to address high-dimensional problems.

- Empirically, extensive experiments on high-dimensional challenging benchmark test functions and practical black-box prompt fine-tuning for large language models demonstrate the effectiveness of our RLTS technique.

## 2 Background

### 2.1 Black-box Optimization

Given a proper function $f(\boldsymbol{x}) : \mathbb{R}^d \to \mathbb{R}$ such that $f(\boldsymbol{x}) > -\infty$, black-box optimization is to minimize $f(\boldsymbol{x})$ by using function queries only. Black-box stochastic optimization methods typically employ a sampling distribution $p(\boldsymbol{x}; \boldsymbol{\theta})$ and optimizes the parameter of the distribution regarding the relaxed problem: $J(\boldsymbol{\theta}) := \mathbb{E}_{p(\boldsymbol{x};\boldsymbol{\theta})}[f(\boldsymbol{x})]$.

Evolution Strategies (ES) [Rechenberg and Eigen, 1973, Nesterov and Spokoiny, 2017] employ a Gaussian distribution $\mathcal{N}(\boldsymbol{\mu}, \sigma^2 \boldsymbol{I})$ for sampling. The approximate gradient descent update is given as

$$\boldsymbol{\mu}_{t+1} = \boldsymbol{\mu}_t - \frac{\beta}{n\sigma} \sum_{i=1}^{n} \boldsymbol{\epsilon}_i f(\boldsymbol{\mu}_t + \sigma \boldsymbol{\epsilon}_i), \tag{1}$$

where $\boldsymbol{\epsilon}_i \sim \mathcal{N}(\boldsymbol{0}, \boldsymbol{I})$ and $\beta$ denotes the step-size. The ES method performs the approximate first-order gradient descent update. As a result, the convergence of ES may be slow. Several second-order gradient descent methods have been proposed to improve convergence. Wierstra et al. [2014a] proposed the natural evolution strategies (NES), which perform the approximate natural gradient update. When a Gaussian distribution $\mathcal{N}(\boldsymbol{\mu}, \boldsymbol{\Sigma})$ is employed for sampling. The update rule of NES is

given in Eq.(2) and Eq.(3):

$$\boldsymbol{\Sigma}_{t+1} = \boldsymbol{\Sigma}_t - \frac{\beta}{n}\sum_{i=1}^{n} f(\boldsymbol{\mu}_t + \boldsymbol{\Sigma}_t^{\frac{1}{2}}\boldsymbol{\epsilon}_i)\left(\boldsymbol{\Sigma}_t^{\frac{1}{2}}\boldsymbol{\epsilon}_i\boldsymbol{\epsilon}_i^{\top}\boldsymbol{\Sigma}_t^{\frac{1}{2}} - \boldsymbol{\Sigma}_t\right) \tag{2}$$

$$\boldsymbol{\mu}_{t+1} = \boldsymbol{\mu}_t - \frac{\beta}{n}\sum_{i=1}^{n} f(\boldsymbol{\mu}_t + \Sigma_t^{\frac{1}{2}}\boldsymbol{\epsilon}_i)\Sigma_t^{\frac{1}{2}}\boldsymbol{\epsilon}_i. \tag{3}$$

where $\boldsymbol{\epsilon}_i \sim \mathcal{N}(\mathbf{0}, \boldsymbol{I})$ and $\boldsymbol{\Sigma}^{\frac{1}{2}} = \boldsymbol{\Sigma}^{\frac{1}{2}\top}$ and $\boldsymbol{\Sigma}^{\frac{1}{2}}\boldsymbol{\Sigma}^{\frac{1}{2}} = \boldsymbol{\Sigma}$. The NES takes advantage of second-order gradient information, which improves the convergence of ES.

Lyu and Tsang [2021] proposed an implicit natural gradient optimizer (INGO) for black-box optimization, which provides an alternative way to compute the natural gradient update. The update rule of INGO is given as in Eq.(4) and Eq.(5):

$$\boldsymbol{\Sigma}_{t+1}^{-1} = \boldsymbol{\Sigma}_t^{-1} + \beta\sum_{i=1}^{n}\frac{f(\boldsymbol{x}_i) - \widehat{\mu}}{n\widehat{\sigma}}\left(\boldsymbol{\Sigma}_t^{-1}(\boldsymbol{x}_i - \boldsymbol{\mu}_t)(\boldsymbol{x}_i - \boldsymbol{\mu}_t)^{\top}\boldsymbol{\Sigma}_t^{-1}\right) \tag{4}$$

$$\boldsymbol{\mu}_{t+1} = \boldsymbol{\mu}_t - \beta\sum_{i=1}^{n}\frac{f(\boldsymbol{x}_i) - \widehat{\mu}}{n\widehat{\sigma}}(\boldsymbol{x}_i - \boldsymbol{\mu}_t). \tag{5}$$

where $\boldsymbol{x}_i \sim \mathcal{N}(\boldsymbol{\mu}_t, \boldsymbol{\Sigma}_t)$, $\widehat{\mu} = \frac{\sum_{i=1}^{n} f(\boldsymbol{x}_i)}{n}$ and $\widehat{\sigma}$ denotes the standard deviation of $f(\boldsymbol{x}_i)$. The normalization $\frac{f(\boldsymbol{x}_i) - \widehat{\mu}}{\widehat{\sigma}}$ is employed to reduce the variance.

CMAES [Hansen, 2006] provides a more sophisticated update rule and performs well on a wide range of black-box optimization problems. All the above stochastic optimization methods rely on sampling. Thus, the sampling phase is vitally important. And a better sampling technique is promising to achieve further improvement.

## 2.2 Rank-1 Lattice

A rank-1 lattice is a particular case of the general lattice with a simple operation for point-set construction. It can be used as Quasi-Monte Carlo for integral approximation [Sloan, 2000, Dick et al., 2013]. A rank-1 lattice point set $\mathcal{P} = \{\boldsymbol{x}_1, \cdots, \boldsymbol{x}_n\}$ can be constructed as Eq.(6):

$$\boldsymbol{x}_i := \frac{i\boldsymbol{z} \bmod n}{n}, i \in \{1, \cdots, n\}, \tag{6}$$

where $\boldsymbol{z} \in \mathbb{Z}^d$ is the so-called generating vector, and mod denotes the modulo operation.

Korobov [1960] proposes a rank-1 lattice with the generating vector having a particular form as Eq.(7)

$$\boldsymbol{z} := [1, k, \cdots, k^{d-1}] \bmod n, \tag{7}$$

where $k$ is searching over $\{1, \cdots, n-1\}$ to reduce approximation error.

Sloan and Reztsov [2002] further proposed a component-by-component searching method for the generating vector without assuming the Korobov form in Eq. (7). Recently, Lyu et al. [2020] proposed a simple closed-form subgroup-based rank-1 lattice by considering the Toroidal distance in the primal lattice space. The generating vector is given as Eq.(8)

$$\boldsymbol{z} = [g^0, g^{\frac{n-1}{2d}}, g^{\frac{2(n-1)}{2d}}, \cdots, g^{\frac{(d-1)(n-1)}{2d}}] \bmod n, \tag{8}$$

where $g$ denotes the primitive root modulo the prime number $n$. More details of the lattice rules for numerical integration can be found in the book [Dick et al., 2022].

In this paper, we proposed a closed-form subgroup rank-1 lattice by ensuring the approximation error terms of the dual lattice have a more regular pattern. In contrast, Lyu et al. [2020] construct the rank-1 lattice evenly spaced in the primal lattice space.

# 3 Fast Rank-1 Lattice Targeted Sampling

## 3.1 Random Rank-1 Lattice Quasi-Monte Carlo Gaussian Sampling

We first show how to construct random rank-1 lattice Quasi-Monte Carlo Gassuain samples. These samples enable us to perform the black-box stochastic optimization listed in section 2.1. More importantly, the nice property of the structure of these samples facilitates a fast targeted sampling.

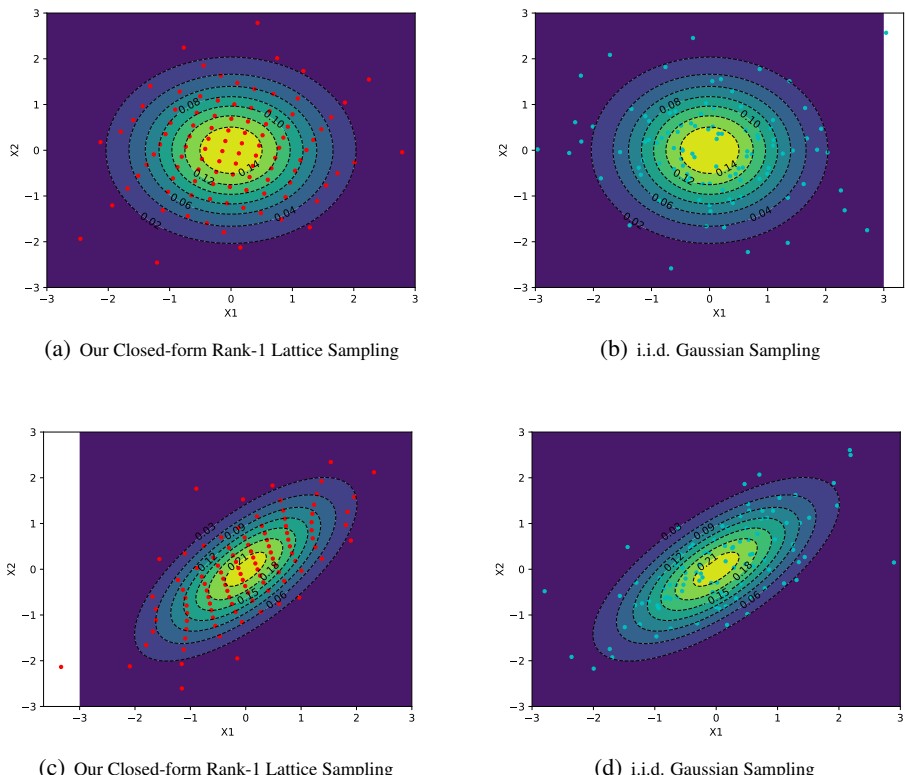

(a) Our Closed-form Rank-1 Lattice Sampling       (b) i.i.d. Gaussian Sampling

(c) Our Closed-form Rank-1 Lattice Sampling       (d) i.i.d. Gaussian Sampling

Figure 1: Illustration of the our closed-form Rank-1 Lattice sampling and i.i.d. Gaussian sampling.

Given a rank-1 lattice point set $\mathcal{P} = \{\boldsymbol{x}_1, \cdots, \boldsymbol{x}_n\}$, we first construct a random shifted rank-1 lattice [Dick et al., 2013] as Eq. (9),

$$\bar{\boldsymbol{x}}_i = \boldsymbol{x}_i + \boldsymbol{\Delta} \bmod 1 \ \ \forall i \in \{1, \cdots, n\}, \tag{9}$$

where $\boldsymbol{\Delta} \sim Uniform[0, 1]^d$, and the mod 1 operation denotes a modulo operation that takes the non-negative fractional part of the input number element-wise. Then, we can construct random QMC Gaussian samples as Eq. (10)

$$\boldsymbol{\epsilon}_i = \Phi^{-1}(\bar{\boldsymbol{x}}_i) \ \ \forall i \in \{1, \cdots, n\}, \tag{10}$$

where $\Phi^{-1}(\cdot)$ computes the inverse cumulative density function of the standard Gaussian distribution w.r.t. the input element-wise. Then, the samples for Gaussian $\mathcal{N}(\boldsymbol{\mu}, \boldsymbol{\Sigma})$ can be constructed as follows:

$$\boldsymbol{\xi}_i = \boldsymbol{\mu} + \boldsymbol{\Sigma}^{\frac{1}{2}} \boldsymbol{\epsilon}_i. \tag{11}$$

An illustration of the random QMC Gaussian samples constructed by our closed-form rank-1 lattice is shown in Figure 1. We can see that our rank-1 lattice QMC Gassuan samples are spaced more evenly w.r.t. the density.

### 3.2 Fast Exact GP Training and Inference with Rank-1 Lattice

This subsection will show how to perform fast exact GP training and inference using our rank-1 lattice samples with a $O(n \log n)$ time complexity w.r.t $n$ samples.

Let $\boldsymbol{K}_\theta$ denotes the kernel Gram matrix, i.e., $\boldsymbol{K}_\theta = [k_\theta(\boldsymbol{x}_i, \boldsymbol{x}_j)]_{1 \le i,j \le n}$, the marginal log-likelihood of a GP model [Williams and Rasmussen, 2006] can be formulated as Eq. (12)

$$\mathcal{L}(p(\boldsymbol{y}|\boldsymbol{X})) = -\frac{1}{2}\boldsymbol{y}^\top(\boldsymbol{K}_\theta + \sigma^2 \boldsymbol{I})^{-1}\boldsymbol{y} - \frac{1}{2}\log(|\boldsymbol{K}_\theta + \sigma^2 \boldsymbol{I}|) - \frac{n}{2}\log 2\pi. \tag{12}$$

The standard GP model needs a $O(n^3)$ time complexity to compute the marginal log-likelihood, which is prohibitive for fast training as an inner step for stochastic optimization.

In this paper, we construct the random QMC samples based on rank-1 lattice, which enables us to perform fast GP training. Specifically, we build the GP model with the rank-1 lattice as the training data instead of the Gaussian samples. Define modulo kernel as Eq. (13):

$$k(\boldsymbol{x}_i, \boldsymbol{x}_j) := k_\Delta(\phi(\boldsymbol{x}_i - \boldsymbol{x}_j)), \tag{13}$$

where $k_\Delta(\cdot)$ is a shift-invariant kernel, and the function $\phi(\boldsymbol{x}_i - \boldsymbol{x}_j)$ is given as Eq. (14)

$$\phi(\boldsymbol{x}_i - \boldsymbol{x}_j) = \min\big((\boldsymbol{x}_i - \boldsymbol{x}_j) \bmod 1, \boldsymbol{1} - (\boldsymbol{x}_i - \boldsymbol{x}_j) \bmod 1\big), \tag{14}$$

where operation $\min(\cdot, \cdot)$ outputs the minimum among its two inputs element-wise, and mod 1 output the positive fractional parts of its inputs element-wise. The nonnegative fractional part of a real number $x$ is $x - \lfloor x \rfloor$, where $\lfloor \cdot \rfloor$ denotes the floor function.

For a GP model with a modulo kernel defined in Eq.(13), the kernel Gram matrix is a circulant matrix thanks to the properties of rank-1 lattice. To be concrete, for rank-1 lattice data, we have Eq.(15)

$$k(\boldsymbol{x}_i, \boldsymbol{x}_j) = k(\boldsymbol{x}_{i+1}, \boldsymbol{x}_{j+1}) = k_\Delta\Big(\min\big(\frac{(i-j)\boldsymbol{z} \bmod n}{n}, \boldsymbol{1} - \frac{(i-j)\boldsymbol{z} \bmod n}{n}\big)\Big). \tag{15}$$

Then the marginal log-likelihood $\mathcal{L}(p(\boldsymbol{y}|\boldsymbol{X}))$ can be computed with a $O(n \log n)$ time complexity by Fast Fourier Transform (FFT).

Specifically, note that the kernel Gram matrix $\boldsymbol{K}_\theta + \sigma^2 \boldsymbol{I}$ is a symmetric circulant matrix generated by vector $\boldsymbol{k}_\Delta$ [1], where $\boldsymbol{k}_\Delta$ is a vector with its $i^{th}$ element given as Eq. (16) .

$$k_{\Delta i} = k_\Delta\Big(\min\big(\frac{(i-1)\boldsymbol{z} \bmod n}{n}, \boldsymbol{1} - \frac{(i-1)\boldsymbol{z} \bmod n}{n}\big)\Big). \tag{16}$$

We know that $\boldsymbol{K}_\theta + \sigma^2 \boldsymbol{I}$ can be diagonalized as $\boldsymbol{K}_\theta + \sigma^2 \boldsymbol{I} = \frac{1}{n} F^* \Lambda F$, where the $j^{th}$ row and $k^{th}$ column element of $F$ is $F_{jk} = e^{-2\pi jk\mathbf{i}/n}$. And the matrix $\Lambda$ is the diagonal eigenvalue matrix that can be computed as $\Lambda = \mathrm{diag}(F\boldsymbol{k}_\Delta)$. The matrix-vector product $F\boldsymbol{k}_\Delta$ can be computed via FFT with $O(n \log n)$ time complexity. And matrix-vector product $\frac{1}{n} F^* \boldsymbol{v}$ for a vector $\boldsymbol{v}$ can be computed via inverse FFT. More details about the properties of circulant matrices and fast computation via FFT can be found in [Gray et al., 2006].

Then, we achieve the fast computation of the terms in log-likelihood as Eq.(17) and Eq.(18):

$$\boldsymbol{y}^\top (\boldsymbol{K}_\theta + \sigma^2 \boldsymbol{I})^{-1} \boldsymbol{y} = \boldsymbol{y}^\top \mathrm{ifft}(\mathrm{fft}(\boldsymbol{y})/\mathrm{fft}(\boldsymbol{k}_\Delta)) \tag{17}$$

$$\log(|\boldsymbol{K}_\theta + \sigma^2 \boldsymbol{I}|) = \sum_{i=1}^n \log(\lambda_i + \sigma^2) = \boldsymbol{1}^\top \log\big(\mathrm{fft}(\boldsymbol{k}_\Delta)\big), \tag{18}$$

where $\mathrm{ifft}(\cdot)$, $\mathrm{fft}(\cdot)$ denotes the inverse FFT and FFT operation, respectively, the operator $/$ in Eq.(17) performs divide element-wise. And the $\log(\cdot)$ is an element-wise operation. And $\lambda_i$ in Eq.(18) denotes the eigenvalue of kernel Gram matrix $\boldsymbol{K}_\theta$.

For inference, GP model has closed-form posterior mean and variance [Williams and Rasmussen, 2006] given as Eq.(19) and Eq.(20) :

$$\widehat{m}(\boldsymbol{x}) = \boldsymbol{k}_\theta(\boldsymbol{x})^\top (\boldsymbol{K}_\theta + \sigma^2 I)^{-1} \boldsymbol{y} \tag{19}$$

$$\widehat{\sigma}^2(\boldsymbol{x}) = k_\theta(\boldsymbol{x}, \boldsymbol{x}) - \boldsymbol{k}_\theta(\boldsymbol{x})^\top (\boldsymbol{K}_\theta + \sigma^2 I)^{-1} \boldsymbol{k}_\theta(\boldsymbol{x}), \tag{20}$$

where $\boldsymbol{k}_\theta(\boldsymbol{x}) = [k_\theta(\boldsymbol{x}, \boldsymbol{x}_1), ..., k_\theta(\boldsymbol{x}, \boldsymbol{x}_n)]^\top$.

With rank-1 lattice input data, we can perform fast inference by Eq.(21) and Eq.(22):

$$\widehat{m}(\boldsymbol{x}) = \boldsymbol{k}_\theta(\boldsymbol{x})^\top \mathrm{ifft}(\mathrm{fft}(\boldsymbol{y})/\mathrm{fft}(\boldsymbol{k}_\Delta)) \tag{21}$$

$$\widehat{\sigma}^2(\boldsymbol{x}) = k_\theta(\boldsymbol{x}, \boldsymbol{x}) - \boldsymbol{k}_\theta(\boldsymbol{x})^\top \mathrm{ifft}(\mathrm{fft}(\boldsymbol{k}_\theta(\boldsymbol{x}))/\mathrm{fft}(\boldsymbol{k}_\Delta)). \tag{22}$$

Both the exact GP training and inference benefit from the structure of rank-1 lattice and FFT acceleration, which can be performed with a $O(n \log n)$ time complexity. A deep learning toolbox, e.g., Pytorch, can be used to train the parameters of the kernel.

---

**Algorithm 1** Fast Coordinate Search

---

**Input:** Number of iterations $T$, weight vector $\boldsymbol{w}$, and generating vector $\boldsymbol{z} = [z_1, \cdots, z_d]$ for rank-1 lattice $\boldsymbol{X}$.
**Initialization:** Initialize $\boldsymbol{x}^*$ by uniformly sampling from grids $\{0, \frac{1}{n}, \cdots, \frac{n-1}{n}\}^d$.
**for** t= 1:T **do**
    **for** q= 1:d **do**
        Compute $\boldsymbol{c}^q = \text{ifft}(\text{fft}(\boldsymbol{k}_\Delta^q(0)) \odot \text{fft}(\widehat{\boldsymbol{k}}_\Delta^q \odot \boldsymbol{w}))$ by Eq.(27).
        Get the index $i^*$ of the minimum elements in $\boldsymbol{c}^q$, and set $x_q^* = \frac{i^* z_q \bmod n}{n}$.
    **end for**
**end for**
**Return:** $\boldsymbol{x}^*$

---

### 3.3 Fast Coordinate Search for Targeted Sampling

This subsection shows how to perform a fast coordinate search for targeted sampling. A rank-1 lattice with $n$ points is contained in a grid $\{0, \frac{1}{n}, \cdots, \frac{n-1}{n}\}^d$. We thus perform a coordinate descent search from the index set $\{0, 1, \cdots, n-1\}^d$ to minimize the GP posterior mean in Eq.(19).

Let $k(\cdot, \cdot) = k_\Delta(\cdot)$ be a shift-invariant kernel with a decomposition structure as Eq. (23):

$$k(\boldsymbol{x}^*, \boldsymbol{x}) = k_\Delta(\phi(\boldsymbol{x}^* - \boldsymbol{x})) = \Pi_{q=1}^d k_\Delta(\phi(x_q^* - x_q)), \tag{23}$$

where $x_q^*, x_q$ denotes the $q^{th}$ element in $\boldsymbol{x}^*, \boldsymbol{x}$, respectively. We can perform a coordinate search by fixing all the components except the $q^{th}$ one as the current working component for index searching. Formally, let $\boldsymbol{w} = (\boldsymbol{K}_\theta + \sigma^2 I)^{-1} \boldsymbol{y}$. Then, we have the GP posterior mean function given as Eq. (24):

$$\widehat{m}(\boldsymbol{x}^*) = \boldsymbol{k}_\Delta^{q\top}(x_q^*)\big(\widehat{\boldsymbol{k}}_\Delta^q \odot \boldsymbol{w}\big), \tag{24}$$

where $\odot$ denotes the element-wise product, and $\boldsymbol{k}_\Delta^q(x_q^*)$ denotes a vector with $i^{th}$ element given as $\boldsymbol{k}_{\Delta i}^q = k_\Delta(\phi(x_q^* - \boldsymbol{X}_{qi}))$, and $\boldsymbol{X}_{qi}$ denotes the element in $q^{th}$-row and $i^{th}$-column of the rank-1 lattice matrix $\boldsymbol{X} = [\boldsymbol{x}_1, \cdots, \boldsymbol{x}_n]$. The vector $\widehat{\boldsymbol{k}}_\Delta^q$ denotes the remainder vector with its $i^{th}$-element given as Eq. (25):

$$\widehat{\boldsymbol{k}}_{\Delta i}^q = \frac{1}{k_\Delta(\phi(x_q^* - \boldsymbol{X}_{qi}))} \Pi_{q=1}^d k_\Delta(\phi(x_q^* - \boldsymbol{X}_{qi})). \tag{25}$$

To optimize the $q^{th}$ component $x_q^*$ of $\boldsymbol{x}^*$, we fix the other components of $\boldsymbol{x}^*$ and the corresponding vector $\widehat{\boldsymbol{k}}_\Delta^q$. We find $x_q^*$ by solving the subproblem given in Eq. (26)

$$x_q^* = \underset{x \in \{0, \cdots, n-1\}}{\arg\min} \ \boldsymbol{k}_\Delta^q(x)^\top \big(\widehat{\boldsymbol{k}}_\Delta^q \odot \boldsymbol{w}\big). \tag{26}$$

Directly enumerate computation of the problem (26) needs a $O(n^2)$ time complexity. In our paper, we can perform a fast computation with $O(n \log n)$ time complexity thanks to the rank-1 lattice $\boldsymbol{X}$. Specially, when $\boldsymbol{X}$ is a rank-1 lattice with the generating vector $\boldsymbol{z} = [z_1, \cdots, z_d]$, then the matrix $\boldsymbol{K}_\Delta^q = [\boldsymbol{k}_\Delta^q(0), \boldsymbol{k}_\Delta^q(\frac{1 z_q \bmod n}{n}), \cdots, \boldsymbol{k}_\Delta^q(\frac{(n-1) z_q \bmod n}{n})]$ forms a circulant matrix, and the problem (26) can be accelerated via FFT by Eq. (27)

$$\boldsymbol{c}^q = \boldsymbol{K}_\Delta^{q\top}\big(\widehat{\boldsymbol{k}}_\Delta^q \odot \boldsymbol{w}\big) = \text{ifft}(\text{fft}(\boldsymbol{k}_\Delta^q(0)) \odot \text{fft}(\widehat{\boldsymbol{k}}_\Delta^q \odot \boldsymbol{w})), \tag{27}$$

where $\text{fft}(\cdot)$ and $\text{ifft}(\cdot)$ denote the FFT and inverse FFT operation. Then, we can achieve $x_q^*$ by the index $i^*$ of the minimum element in vector $\boldsymbol{c}^q = \boldsymbol{K}_\Delta^{q\top}\big(\widehat{\boldsymbol{k}}_\Delta^q \odot \boldsymbol{w}\big)$, and set $x_q^* = \frac{i^* z_q \bmod n}{n}$.

We present the algorithm of the fast coordinate search in Algorithm 1. The Algorithm 1 return a targeted sample with a small prediction value in a fast manner. We can use the targeted sample to accelerate the stochastic optimization. Finally, we present our overall stochastic optimization algorithm in the Algorithm 2. We choose INGO [Lyu and Tsang, 2021] as our backbone algorithm because of its simple implementation and fewer hyperparameters. One can plug our RLTS into other stochastic optimization methods to improve query efficiency.

---

[1]The first element of $\boldsymbol{k}_\Delta$ is set to $k_\Delta(\boldsymbol{0}) + \sigma^2$.

---

**Algorithm 2** Rank-1 Lattice Targeted Sampling

---

**Input:** Number of batch samples $n$, step-size $\beta$ and $\eta$, number of internal iterations $T$ for Fast Coordinate Search, and initial variance $\sigma^2$.

**Initialization:** Initialize $\boldsymbol{\mu}_0 = \mathbf{0}$ and $\Sigma_0 = \sigma^2 \boldsymbol{I}$.

**while** Termination condition not satisfied **do**

    Sample a shift vector $\Delta$ uniformly from $[0,1]^d$.

    Construct shifted rank-1 lattice $\bar{\boldsymbol{X}} = [\bar{\boldsymbol{x}}_1, \cdots, \bar{\boldsymbol{x}}_n]$ by Eq.(9).

    Construct QMC Gaussian Samples $\boldsymbol{\epsilon}_1, \cdots, \boldsymbol{\epsilon}_n$ by Eq.(10).

    Set $\boldsymbol{\xi}_i = \boldsymbol{\mu}_t + \Sigma_t^{\frac{1}{2}} \boldsymbol{\epsilon}_i$ for $i \in \{1, \cdots n\}$.

    Query the batch observations $\{f(\boldsymbol{\xi}_1), ..., f(\boldsymbol{\xi}_n)\}$

    Compute $\widehat{\sigma} = \mathrm{std}(f(\boldsymbol{\xi}_1), ..., f(\boldsymbol{\xi}_n))$.

    Compute $\widehat{\mu} = \frac{1}{n} \sum_{i=1}^n f(\boldsymbol{\xi}_i)$.

    Set $y_i = \frac{f(\boldsymbol{\xi}_i) - \widehat{\mu}}{\widehat{\sigma}}$ for $i \in \{1, \cdots n\}$.

    Perform fast exact GP training with rank-1 lattice $\bar{X}$ and $\boldsymbol{y}$ by Eq.(17) and Eq.(18).

    Get targeted grid sample $\bar{\boldsymbol{x}}^*$ by Algorithm 1 with $T$ steps.

    Get targeted Gaussian sample $\boldsymbol{\xi}^* = \Phi^{-1}(\bar{\boldsymbol{x}}^* + \Delta \bmod 1)$

    Query the observation $f(\boldsymbol{\xi}^*)$.

    Set $\Sigma_{t+1}^{-1} = \Sigma_t^{-1} + \frac{\beta}{n} \sum_{i=1}^n y_i \Sigma_t^{-\frac{1}{2}} \boldsymbol{\epsilon}_i \boldsymbol{\epsilon}_i^\top \Sigma_t^{-\frac{1}{2}}$.

    Set $\boldsymbol{\mu}_{t+1} = \boldsymbol{\mu}_t - \frac{\beta}{n} \sum_{i=1}^n y_i \Sigma_t^{\frac{1}{2}} \boldsymbol{\epsilon}_i$

    **if** $f(\boldsymbol{\xi}^*) < \min_{i \in \{1, \cdots, n\}} f(\boldsymbol{\xi}_i)$ **then**

        Set $\boldsymbol{\mu}_{t+1} = (1 - \eta)\boldsymbol{\mu}_{t+1} + \eta \boldsymbol{\xi}^*$

    **end if**

**end while**

---

### 3.4 Closed-form Rank-1 Lattice Construction

This subsection will show how to construct our closed-form rank-1 lattice for fast sampling. For $\forall \boldsymbol{x}, \boldsymbol{y} \in [0,1]^d$ and $\alpha > 1$, define a reproducing kernel as Eq. (28)

$$K(\boldsymbol{x}, \boldsymbol{y}) = \sum_{\boldsymbol{k} \in \mathbb{Z}^d} \gamma_\alpha(\boldsymbol{k}) \exp\left(2\pi \mathbf{i} \boldsymbol{k}^\top (\boldsymbol{x} - \boldsymbol{y})\right), \tag{28}$$

where $\mathbf{i}^2 = -1$ and $\gamma_\alpha(\boldsymbol{k}) = \prod_{j=1}^d \gamma_\alpha(k_j)$ with $\gamma_\alpha(k)$ is given as follows:

$$\gamma_\alpha(k) = \begin{cases} 1 & \text{if } k = 0 \\ |k|^{-\alpha} & \text{if } k \neq 0. \end{cases} \tag{29}$$

A Korobov space is a reproducing kernel Hilbert space (RKHS) associated with the kernel in Eq.(28), denoted as $\mathcal{H}_k$.

Our closed form of the generating vector is given as Eq.(30):

$$\boldsymbol{z} = [g^0, g^{\frac{n-1}{2d-1}}, g^{\frac{2(n-1)}{2d-1}}, \cdots, g^{\frac{(d-1)(n-1)}{2d-1}}] \bmod n, \tag{30}$$

where $g$ denotes the primitive root modulo the prime number $n$, and $(2d-1)|(n-1)$. Then, our close-form rank-1 lattice can be achieved by Eq. (6)

Given a point set $\mathcal{P} = \{\boldsymbol{x}_1, \cdots, \boldsymbol{x}_n\}$, the square worst case integral approximation error for $f \in \mathcal{H}_k$ is defined as Eq.(31):

$$e^2(\mathcal{H}_k; \mathcal{P}) = \sup_{f \in \mathcal{H}_k, \|f\|_{\mathcal{H}_k} \leq 1} \left| \int_{[0,1]^d} f(\boldsymbol{x}) \mathrm{d}\boldsymbol{x} - \frac{1}{n} \sum_{j=0}^{n-1} f(\boldsymbol{x}_j) \right|^2. \tag{31}$$

We further show that our rank-1 lattice constructed by Eq. (30) has a regular worst-case error pattern in Theorem 1. The proof is given in the Appendix.

**Theorem 1.** *Let $n$ be a prime number such that $(2d - 1)|(n - 1)$. Suppose the integrand function $f \in \mathcal{H}_k, \|f\|_{\mathcal{H}_k} \leq 1$, the square worst-case integral approximation error of rank-1 lattice $\mathcal{P}$ constructed by Eq.(30) is given as Eq.(32):*

$$e^2(\mathcal{H}_k; \mathcal{P}) = \frac{1}{2n} \mathbf{1}^\top \left( \boldsymbol{h}^0 \odot \cdots \odot \boldsymbol{h}^{2d-2} - \mathbf{1} - (\boldsymbol{h}^1 \odot \cdots \odot \boldsymbol{h}^{d-1} - \mathbf{1}) \odot \boldsymbol{h}^0 \odot (\boldsymbol{h}^{-1} \odot \cdots \odot \boldsymbol{h}^{-(d-1)} - \mathbf{1}) \right) + \frac{1}{n^\alpha} \zeta(\alpha, 1), \tag{32}$$

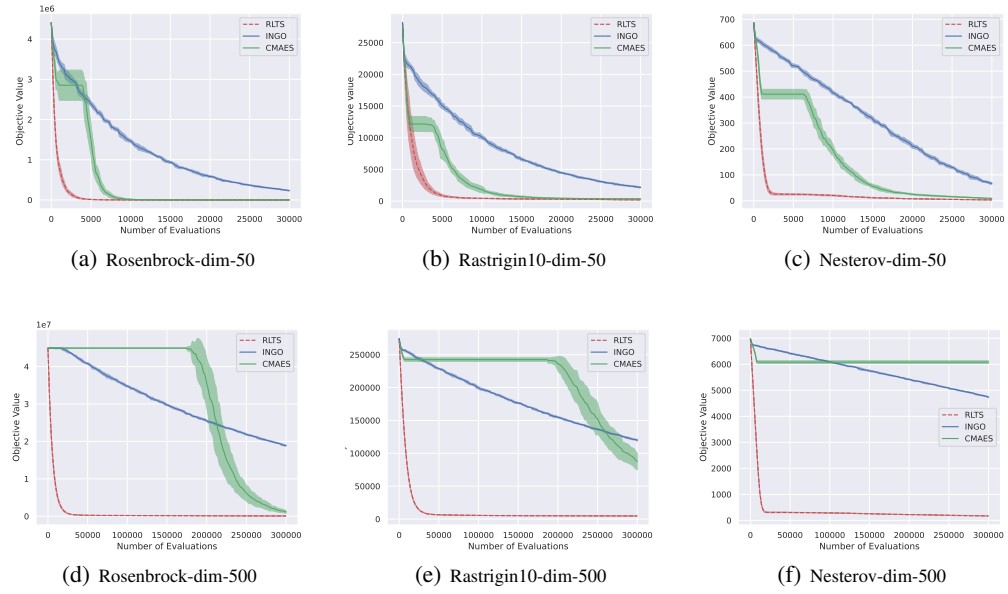

Figure 2: Cumulative min objective value v.s. the number of query evaluations on 50-dimensional and 500-dimensional benchmark test functions.

where $\odot$ denotes the element-wise product, symbol $\mathbf{1}$ denotes the vector with elements all ones, and $\boldsymbol{h}^i = \boldsymbol{F}^i \boldsymbol{\gamma}$ with $\boldsymbol{F}$ as the discrete Fourier matrix, i.e., $\boldsymbol{F}_{jk} = \exp(2\pi \mathbf{i} \frac{jk}{n})$, and $\boldsymbol{F}^i$ denotes the matrix after permutation of the rows of $\boldsymbol{F}$ such that the $j^{th}$ row of $\boldsymbol{F}^i$ equals to the $\widetilde{j}^{th}$ row of $\boldsymbol{F}$, where $\widetilde{j} = jg^{\frac{i(n-1)}{2d-1}} \mod n$. And $\boldsymbol{\gamma} = [\gamma_1, \cdots, \gamma_n]^\top$ with $\gamma_k = \frac{1}{n^\alpha}\left(\zeta(\alpha, \frac{k_i}{n}) + \zeta(\alpha, \frac{n-k_i}{n})\right)$ for $k \in \{1, \cdots, n-1\}$ and $\gamma_n = 1 + \frac{2}{n^\alpha}\zeta(\alpha, 1)$, where $\zeta(\cdot, \cdot)$ denotes the Hurwitz zeta function.

**Remarks:** The term $H = \boldsymbol{h}^0 \odot \cdots \odot \boldsymbol{h}^{2d-2} - \mathbf{1} - (\boldsymbol{h}^1 \odot \cdots \odot \boldsymbol{h}^{d-1} - \mathbf{1}) \odot \boldsymbol{h}^0 \odot (\boldsymbol{h}^{-1} \odot \cdots \odot \boldsymbol{h}^{-(d-1)} - \mathbf{1})$ has a regular pattern because of $\{g^0, g^{\frac{n-1}{2d-1}}, g^{\frac{2(n-1)}{2d-1}}, \cdots, g^{\frac{(d-1)(n-1)}{2d-1}}, \cdots, g^{\frac{(2d-2)(n-1)}{2d-1}}\} \mod n$ forms a subgroup of $\{1, \cdots, n-1\} \mod n$. According to the Lagrange's theorem in group theory [Dummit and Foote, 2004], the vector $\boldsymbol{h}^0 \odot \cdots \odot \boldsymbol{h}^{2d-2}$ has $\frac{n-1}{2d-1}$ different elements.

## 4 Experiments

We replace the i.i.d. Gaussian sampling of the INGO [Lyu and Tsang, 2021] with our RLTS. We evaluate our RLTS by comparing it with the standard INGO and the CMAES [Hansen, 2006]. In all the experiments, we keep the number of batch samples and the initialization the same for RLTS, INGO and CMAES. For all the methods, we initialize the $\boldsymbol{\mu} = \mathbf{0}$. For INGO and RLTS, we set the step-size parameter $\beta = 0.2$ in all experiments. For RLTS, we set the parameter $\eta = 1$ in all experiments.

### 4.1 Evaluation on Benchmark Functions

We first evaluate our RLTS on challenging benchmark test functions: Rosenbrock, Rastrigin, and Nesterov. Rastrigin and Rosenbrock are smooth multi-mode functions, and Nesterov is a non-smooth function. These functions are very challenging benchmarks for black-box optimization. We offset the optimum by setting $\boldsymbol{x} = \boldsymbol{x} - 5$ of the test functions. This increases the distance between the optimum and the initial point $\boldsymbol{\mu} = \mathbf{0}$, which makes the test problems more challenging. We implement INGO by ourselves. For CMAES, we use the publicly available code [2]. We initialized $\boldsymbol{\Sigma} = \boldsymbol{I}$ for all the methods.

---

[2]https://pypi.org/project/cma/

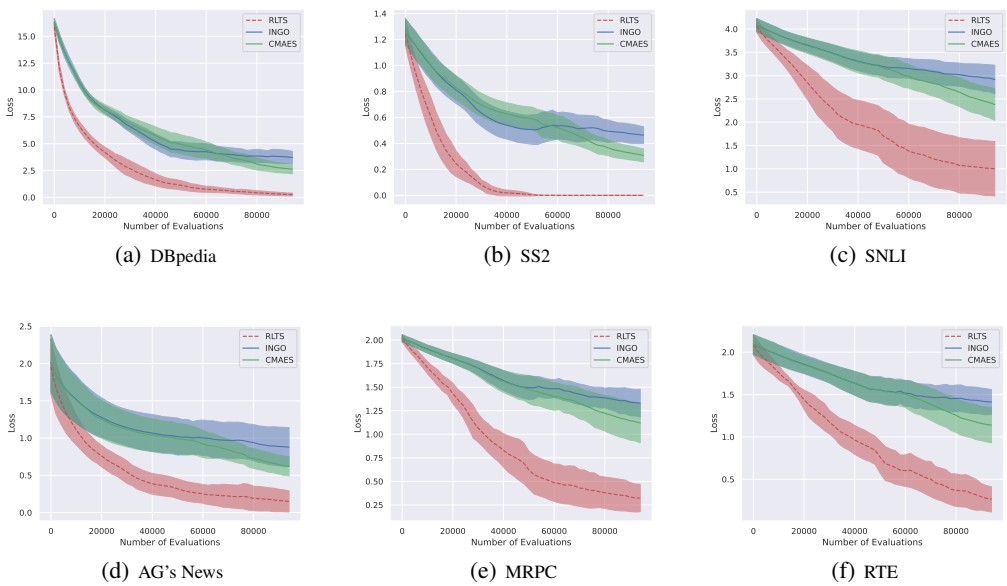

Figure 3: Hinge loss v.s. the number of query evaluations on different black-box fine-tuning models.

We evaluate RLTS on 50 and 500-dimensional problems. The batchsize of all the methods are set to 200 and 2000 for 50 and 500-dimensional problems, respectively. All the experiments are performed in ten independent runs. The experimental results are shown in Figure 2. From Figure 2, we can observe that RLTS consistently converge faster than INGO on all the test functions on both 50-dimensional and 500-dimensional cases. It shows that our RLTS significantly improves the query efficiency of INGO, which verifies the effectiveness of RLTS. Moreover, we can see that RLTS outperforms CMAES on all the test functions on both 50-dimensional and 500-dimensional cases. In addition, we see that CMAES converge slowly on the 500-dimensional benchmark problems, while RLTS converges faster.

## 4.2 Evaluation on Black-box Prompt Fine-tuning Tasks

Prompt fine-tuning of large language models is a promising direction to achieve expertise models efficiently for downstream tasks. We evaluate our RLTS on black-box prompt fine-tuning tasks.

We employ the deep model in [Sun et al., 2022a] with publicly available code [3] as the backbone model for black-box prompt fine-tuning. It has 24 layers. For each layer, we set the dimension of the continuous prompt to 500. Thus, the total dimension is $24 \times 500$. We employ the hinge loss of training data as the black-box objective. Six benchmark datasets for different language tasks are employed for evaluation: DBpedia, SS2, SNLI, AG's News, MRPC and RTE. The SST2 [Socher et al., 2013] dataset is a dataset for the sentiment analysis task. AG's News and DBPedia datasets [Zhang et al., 2015] are used for topic classification tasks. SNLI [Bowman et al., 2015] and RTE [Wang et al., 2019] are employed for natural language inference. MRPC dataset [Dolan and Brockett, 2005] is used for the paraphrasing task.

In all the experiments, we keep the number of batch samples and the initialization the same for RLTS, INGO and CMAES. We set the number of batch samples to 2000. Specifically, our RLTS employs 1999 rank-1 lattice QMC Gaussian samples and one sample from targeted sampling. INGO employs 1999 rank-1 lattice QMC Gaussian samples and one Gaussian sample. CMAES employs 2000 Gaussian samples. We initialize the $\boldsymbol{\mu} = \boldsymbol{0}$ and $\boldsymbol{\Sigma} = 0.2\boldsymbol{I}$ for all the methods. For INGO and RLTS, we set the step-size parameter $\beta = 0.2$ in all experiments. For RLTS, we set the parameter $\eta = 1$ in all experiments. All the experiments are performed in five independent runs with seeds in $\{1, 2, 3, 4, 5\}$. The layer-wise coordinate descent update approach in [Sun et al., 2022a] is employed for all the methods.

---

[3]https://github.com/txsun1997/Black-Box-Tuning

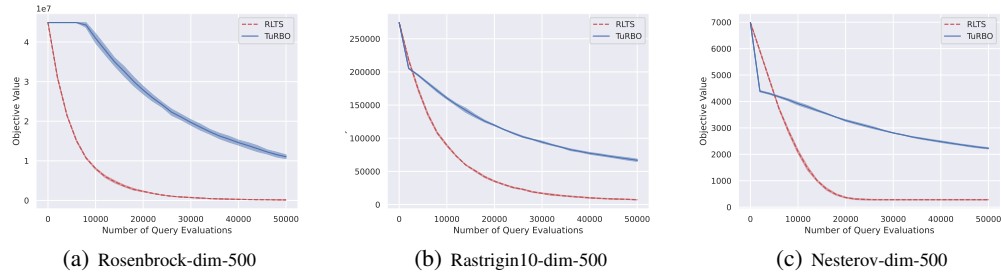

| (a) Rosenbrock-dim-500 | (b) Rastrigin10-dim-500 | (c) Nesterov-dim-500 |

Figure 4: Cumulative min objective value v.s. the number of query evaluations on 500-dimensional benchmark test functions.

The experimental results of mean objective $\pm$ std v.s. the number of queries are shown in Figure 3. From Figure 3, we can observe that our RLTS decreases the objective significantly faster than INGO and CMAES on all six fine-tuning tasks, which shows the superior query efficiency of our RLTS.

### 4.3 Additional Comparison with High-dimensional Bayesian Optimization

We further compare our RLTS with the high-dimensional BO method TuRBO [Eriksson et al., 2019]. We evaluate RLTS on the three benchmark functions: the Rosenbrock function, the Rastrigin10 function, and the Nesterov function. We offset the optimum by setting $x = x - 5$ of the test functions. The dimension is set to 500. The number of initial points of TuRBO is set to 2000. The batch size of both RLTS and TuRBO is set to 2000. The maximum number of queries is set to 50,000. We employ the default box boundary for TuRBO, i.e., $[-5, 10]^d$. The initial parameter $\boldsymbol{\mu}$ of RLTS is set to $\boldsymbol{\mu} = \mathbf{0}$, and $\boldsymbol{\Sigma}$ is set to $\boldsymbol{\Sigma} = \boldsymbol{I}$. For TuRBO, we employ the official code provided in the paper [Eriksson et al., 2019]. All the methods are performed in three independent runs.

The convergence performance regarding the number of query evaluations is shown in Figure 4. We can observe that RLTS converges faster than TuRBO on the benchmark test problems, demonstrating that RLTS improves query efficiency.

We further report the running time of RLTS and TuRBO on the same machine for evaluation. The results are shown in Table 1. We can observe that RLTS performs significantly faster than TuRBO, achieving around 300 times speedup regarding running time. The computation time of Bayesian Optimization usually grows cubically fast as the number of queries increases. In contrast, our RLTS reduces the expensive $O(n^3)$ operation to $O(n \log n)$ time complexity, which enables a fast plug-in of the ES-type algorithms.

Table 1: Running time on benchmark test functions. Symbol (s) denotes seconds.

|  | Rosenbrock | Rastrigin10 | Nesterov |
|---|---|---|---|
| RLTS | 83.62(s) | 84.04(s) | 83.46(s) |
| TuRBO | 25927.39(s) | 25941.66(s) | 25697.87(s) |

## 5 Conclusion

We proposed a novel Rank-1 Lattice Targeted Sampling technique in this paper. Our RLTS has a $O(n \log n)$ time complexity w.r.t. $n$ batch samples, which is fast for plugging into stochastic optimization methods to improve query efficiency while scaling up to high-dimensional problems. Empirically, we plugged our RLTS into the sampling phase of INGO, significantly improving the query efficiency on benchmark test functions and black-box prompt fine-tuning tasks. Moreover, we proposed a closed-form rank-1 lattice by analyzing the integral approximation error of functions in Korobov space. Our closed-form rank-1 lattice provides an efficient way for QMC Gaussian sampling, with properties enabling fast exact GP training and inference with a $O(n \log n)$ time complexity, which is critical for our RLTS to be a fast internal step for stochastic optimization. In addition, our closed-form rank-1 lattice is a fundamental tool that may have potential applications beyond the black-box optimization task.

## Acknowledgement

We thank the anonymous reviewers for their valuable comments and helpful suggestions.

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

# Appendix

## A   Proof of Theorem 1

**Theorem 1.** *Let $n$ be a prime number such that $(2d-1)|(n-1)$. Suppose the integrand function $f \in \mathcal{H}_k, \|f\|_{\mathcal{H}_k} \leq 1$, the square worst-case integral approximation error of rank-1 lattice $\mathcal{P}$ constructed by Eq.(30) is given as Eq.(33):*

$$e^2(\mathcal{H}_k; \mathcal{P}) = \frac{1}{2n} \mathbf{1}^\top \left( \mathbf{h}^0 \odot \cdots \odot \mathbf{h}^{2d-2} - \mathbf{1} - (\mathbf{h}^1 \odot \cdots \odot \mathbf{h}^{d-1} - \mathbf{1}) \odot \mathbf{h}^0 \odot (\mathbf{h}^{-1} \odot \cdots \odot \mathbf{h}^{-(d-1)} - \mathbf{1}) \right) + \frac{1}{n^\alpha} \zeta(\alpha, 1),$$
(33)

*where $\odot$ denotes the element-wise product, symbol $\mathbf{1}$ denotes the vector with elements all ones, and $\mathbf{h}^i = \mathbf{F}^i \boldsymbol{\gamma}$ with $\mathbf{F}$ as the discrete Fourier matrix, i.e., $\mathbf{F}_{jk} = \exp(2\pi i \frac{jk}{n})$, and $\mathbf{F}^i$ denotes the matrix after permutation of the rows of $\mathbf{F}$ such that the $j^{th}$ row of $\mathbf{F}^i$ equals to the $\widetilde{j}^{th}$ row of $\mathbf{F}$, where $\widetilde{j} = jg^{\frac{i(n-1)}{2d-1}} \mod n$. And $\boldsymbol{\gamma} = [\gamma_1, \cdots, \gamma_n]^\top$ with $\gamma_k = \frac{1}{n^\alpha}\left( \zeta(\alpha, \frac{k_i}{n}) + \zeta(\alpha, \frac{n-k_i}{n}) \right)$ for $k \in \{1, \cdots, n-1\}$ and $\gamma_n = 1 + \frac{2}{n^\alpha}\zeta(\alpha, 1)$, where $\zeta(\cdot, \cdot)$ denotes the Hurwitz zeta function.*

To prove our main Theorem 1, we begin with several Lemma.

**Lemma 1.** *For $\forall \boldsymbol{x}, \boldsymbol{y} \in [0,1]^d$ and $\alpha > 1$, define a reproducing kernel as Eq.(34)*

$$K(\boldsymbol{x}, \boldsymbol{y}) = \sum_{\boldsymbol{k} \in \mathbb{Z}^d} \gamma_\alpha(\boldsymbol{k}) \exp\left( 2\pi i \boldsymbol{k}^\top (\boldsymbol{x} - \boldsymbol{y}) \right),$$
(34)

*where $\gamma_\alpha(\boldsymbol{k}) = \prod_{j=1}^d \gamma_\alpha(k_j)$ with $\gamma_\alpha(k)$ given in Eq.(35)*

$$\gamma_\alpha(k) = \begin{cases} 1 & \text{if } k = 0 \\ |k|^{-\alpha} & \text{if } k \neq 0. \end{cases}$$
(35)

*Let $\mathcal{P} = [\boldsymbol{x}_1, \cdots, \boldsymbol{x}_n]$ be a rank-1 lattice constructed by the generating vector $\boldsymbol{z}$ with a prime number $n$. Then, for $\forall f \in \mathcal{H}_k, \|f\|_{\mathcal{H}_k} \leq 1$ associated with the reproducing kernel Eq.(34), we have the square worst-case integral approximation error of $\mathcal{P}$ as Eq.(36).*

$$e^2(\mathcal{H}_k; \mathcal{P}) = \sup_{f \in \mathcal{H}_k, \|f\|_{\mathcal{H}_k} \leq 1} \left| \int_{[0,1]^d} f(\boldsymbol{x}) d\boldsymbol{x} - \frac{1}{n} \sum_{j=0}^{n-1} f(\boldsymbol{x}_j) \right|^2 = \sum_{\boldsymbol{k} \in L^\perp \setminus \{\mathbf{0}\}} \gamma_\alpha(\boldsymbol{k})$$
(36)

*where $L^\perp$ denote the dual lattice defined in Eq.(37).*

$$L^\perp := \{ \boldsymbol{k} | \boldsymbol{k}^\top \boldsymbol{z} \equiv 0 \ (mod \ n), \boldsymbol{k} \in \mathbb{Z}^d \}.$$
(37)

*Proof.* Given a point set $\mathcal{P} = \{\boldsymbol{x}_1, \cdots, \boldsymbol{x}_n\}$, the worst case approximation error for $\forall f \in \mathcal{H}_k, \|f\|_{\mathcal{H}_k} \leq 1$ is

$$e^2(\mathcal{H}_k; \mathcal{P}) = \sup_{f \in \mathcal{H}_k, \|f\|_{\mathcal{H}_k} \leq 1} \left| \int_{[0,1]^d} f(\boldsymbol{x}) d\boldsymbol{x} - \frac{1}{n} \sum_{j=0}^{n-1} f(\boldsymbol{x}_j) \right|^2$$
(38)

$$= \sup_{f \in \mathcal{H}_k, \|f\|_{\mathcal{H}_k} \leq 1} \left| \left\langle f, \int_{[0,1]^d} K(\boldsymbol{x}, \cdot) d\boldsymbol{x} - \frac{1}{n} \sum_{j=0}^{n-1} K(\boldsymbol{x}_j, \cdot) \right\rangle_{\mathcal{H}_k} \right|^2$$
(39)

$$= \sup_{f \in \mathcal{H}_k, \|f\|_{\mathcal{H}_k} \leq 1} \|f\|_{\mathcal{H}_k} \left\| \int_{[0,1]^d} K(\boldsymbol{x}, \cdot) d\boldsymbol{x} - \frac{1}{n} \sum_{j=0}^{n-1} K(\boldsymbol{x}_j, \cdot) \right\|_{\mathcal{H}_k}$$
(40)

$$= \int_{[0,1]^d} \int_{[0,1]^d} K(\boldsymbol{x}, \boldsymbol{y}) d\boldsymbol{x} d\boldsymbol{y} - \frac{2}{n} \sum_{j=1}^n \int_{[0,1]^d} K(\boldsymbol{x}, \boldsymbol{x}_j) d\boldsymbol{x} + \frac{1}{n^2} \sum_{i,j=1}^n K(\boldsymbol{x}_i, \boldsymbol{x}_j)$$
(41)

Then, from the definition of the reproducing kernel $K(\boldsymbol{x}, \boldsymbol{y})$ in Eq.(34), we know that

$$\int_{[0,1]^d} \int_{[0,1]^d} K(\boldsymbol{x}, \boldsymbol{y}) \mathrm{d}\boldsymbol{x}\mathrm{d}\boldsymbol{y} = \int_{[0,1]^d} \int_{[0,1]^d} \sum_{\boldsymbol{k}\in\mathbb{Z}^d} \gamma_\alpha(\boldsymbol{k}) \exp\left(2\pi\mathbf{i}\boldsymbol{k}^\top(\boldsymbol{x}-\boldsymbol{y})\right) \mathrm{d}\boldsymbol{x}\mathrm{d}\boldsymbol{y} \tag{42}$$

$$= 1 + \sum_{\boldsymbol{k}\in\mathbb{Z}^d, \boldsymbol{k}\neq\boldsymbol{0}} \gamma_\alpha(\boldsymbol{k}) \int_{[0,1]^d} \int_{[0,1]^d} \exp\left(2\pi\mathbf{i}\boldsymbol{k}^\top(\boldsymbol{x}-\boldsymbol{y})\right) \mathrm{d}\boldsymbol{x}\mathrm{d}\boldsymbol{y} \tag{43}$$

$$= 1 + \sum_{\boldsymbol{k}\in\mathbb{Z}^d, \boldsymbol{k}\neq\boldsymbol{0}} \gamma_\alpha(\boldsymbol{k}) \cdot 0 = 1 \tag{44}$$

In addition, the second term in Eq.(41) as follows

$$-\frac{2}{n} \sum_{j=1}^{n} \int_{[0,1]^d} K(\boldsymbol{x}, \boldsymbol{x}_j) \mathrm{d}\boldsymbol{x} \tag{45}$$

$$= -\frac{2}{n} \sum_{j=1}^{n} \int_{[0,1]^d} \sum_{\boldsymbol{k}\in\mathbb{Z}^d} \gamma_\alpha(\boldsymbol{k}) \exp\left(2\pi\mathbf{i}\boldsymbol{k}^\top(\boldsymbol{x}-\boldsymbol{x}_j)\right) \mathrm{d}\boldsymbol{x} \tag{46}$$

$$= -\frac{2}{n} \sum_{j=1}^{n} \gamma_\alpha(\boldsymbol{0}) - \frac{2}{n} \sum_{j=1}^{n} \sum_{\boldsymbol{k}\in\mathbb{Z}^d, \boldsymbol{k}\neq\boldsymbol{0}} \gamma_\alpha(\boldsymbol{k}) \int_{[0,1]^d} \exp\left(2\pi\mathbf{i}\boldsymbol{k}^\top(\boldsymbol{x}-\boldsymbol{x}_j)\right) \mathrm{d}\boldsymbol{x} \tag{47}$$

$$= -\frac{2}{n} \sum_{j=1}^{n} \gamma_\alpha(\boldsymbol{0}) - \frac{2}{n} \sum_{j=1}^{n} \sum_{\boldsymbol{k}\in\mathbb{Z}^d, \boldsymbol{k}\neq\boldsymbol{0}} \gamma_\alpha(\boldsymbol{k}) \cdot 0 \tag{48}$$

$$= -2 \tag{49}$$

Moreover, from the definition of rank-1 lattice $\mathcal{P}$ with prime $n$ and generating vector $\boldsymbol{z}$, we have the third term in Eq.(41) as follows

$$\frac{1}{n^2} \sum_{i,j=1}^{n} K(\boldsymbol{x}_i, \boldsymbol{x}_j) \tag{50}$$

$$= \frac{1}{n^2} \sum_{i,j=1}^{n} \sum_{\boldsymbol{k}\in\mathbb{Z}^d} \gamma_\alpha(\boldsymbol{k}) \exp\left(2\pi\mathbf{i}\boldsymbol{k}^\top(\boldsymbol{x}_i-\boldsymbol{x}_j)\right) \tag{51}$$

$$= 1 + \frac{1}{n^2} \sum_{i,j=1}^{n} \sum_{\boldsymbol{k}\in\mathbb{Z}^d, \boldsymbol{k}\neq\boldsymbol{0}} \gamma_\alpha(\boldsymbol{k}) \exp\left(\frac{2\pi\mathbf{i}(i-j)\boldsymbol{k}^\top\boldsymbol{z}}{n}\right) \tag{52}$$

$$= 1 + \sum_{\boldsymbol{k}\in\mathbb{Z}^d, \boldsymbol{k}\neq\boldsymbol{0}} \gamma_\alpha(\boldsymbol{k}) \frac{1}{n^2} \sum_{i,j=1}^{n} \exp\left(\frac{2\pi\mathbf{i}(i-j)\boldsymbol{k}^\top\boldsymbol{z}}{n}\right) \tag{53}$$

$$= 1 + \sum_{\boldsymbol{k}\in\mathbb{Z}^d, \boldsymbol{k}\neq\boldsymbol{0}} \gamma_\alpha(\boldsymbol{k}) \frac{1}{n} \sum_{j=1}^{n} \exp\left(\frac{2\pi\mathbf{i}j\boldsymbol{k}^\top\boldsymbol{z}}{n}\right) \tag{54}$$

Put Eq.(44), Eq.(49) and Eq.(54) together , we know that

$$e^2(\mathcal{H}_k; \mathcal{P}) = \sum_{\boldsymbol{k}\in\mathbb{Z}^d, \boldsymbol{k}\neq\boldsymbol{0}} \gamma_\alpha(\boldsymbol{k}) \frac{1}{n} \sum_{j=1}^{n} \exp\left(\frac{2\pi\mathbf{i}j\boldsymbol{k}^\top\boldsymbol{z}}{n}\right) \tag{55}$$

Note that for a prime number $n$, we have

$$\frac{1}{n} \sum_{j=1}^{n} \exp\left(\frac{2\pi\mathbf{i}j\boldsymbol{k}^\top\boldsymbol{z}}{n}\right) = \begin{cases} 1 & \text{if } \boldsymbol{k}^\top\boldsymbol{z} \equiv 0 \bmod n \\ 0 & \text{otherwise} \end{cases} \tag{56}$$

It follows that

$$e^2(\mathcal{H}_k; \mathcal{P}) = \sup_{f\in\mathcal{H}_k, \|f\|_{\mathcal{H}_k}\leq 1} \left| \int_{[0,1]^d} f(\boldsymbol{x})\mathrm{d}\boldsymbol{x} - \frac{1}{n} \sum_{j=0}^{n-1} f(\boldsymbol{x}_j) \right|^2 = \sum_{\boldsymbol{k}\in L^\perp\setminus\{\boldsymbol{0}\}} \gamma_\alpha(\boldsymbol{k}), \tag{57}$$

where $L^\perp := \{ \boldsymbol{k} | \boldsymbol{k}^\top \boldsymbol{z} \equiv 0 \pmod{n}, \boldsymbol{k} \in \mathbb{Z}^d \}$ denotes the dual lattice.

$\square$

**Lemma 2.** *Given a prime $n$, construct a rank-1 lattice $\mathcal{P} = [\boldsymbol{x}_1, \cdots, \boldsymbol{x}_n]$ by the generating vector $\boldsymbol{z} = [z_1, \cdots, z_d]$, then we have that*

$$e^2(\mathcal{H}_k; \mathcal{P}) = -1 + \frac{1}{n} \sum_{j=0}^{n-1} \prod_{i=1}^{d} \Big( \sum_{k_i \in \{1, \cdots, n\}} \chi(k_i) \exp\Big(2\pi\mathbf{i}\frac{k_i j z_i}{n}\Big) \Big), \tag{58}$$

*where function $\chi(\cdot)$ on domain $\{1, \cdots, n\}$ is given as Eq.(59)*

$$\chi(k_i) = \begin{cases} 1 + \frac{2}{n^\alpha}\zeta(\alpha, 1) & \text{if } k_i = n \\ \frac{1}{n^\alpha}\big(\zeta(\alpha, \frac{k_i}{n}) + \zeta(\alpha, \frac{n-k_i}{n})\big) & \text{otherwise} \end{cases}, \tag{59}$$

*where $\zeta(\cdot, \cdot)$ denotes the Hurwitz zeta function.*

*Proof.* From Lemma 1, we know that

$$e^2(\mathcal{H}_k; \mathcal{P}) = \sum_{\boldsymbol{k} \in \mathbb{Z}^d \setminus \{\boldsymbol{0}\}} \gamma_\alpha(\boldsymbol{k}) \left( \frac{1}{n} \sum_{j=0}^{n-1} \exp\Big(2\pi\mathbf{i}\frac{\boldsymbol{k}^\top \boldsymbol{x}_j}{n}\Big) \right) \tag{60}$$

$$= -1 + \frac{1}{n} \sum_{j=0}^{n-1} \sum_{\boldsymbol{k} \in \mathbb{Z}^d} \gamma_\alpha(\boldsymbol{k}) \exp\Big(2\pi\mathbf{i}\frac{\boldsymbol{k}^\top \boldsymbol{x}_j}{n}\Big) \tag{61}$$

$$= -1 + \frac{1}{n} \sum_{j=0}^{n-1} \prod_{i=1}^{d} \Big( \sum_{k_i \in \mathbb{Z}} \gamma_\alpha(k_i) \exp\Big(2\pi\mathbf{i}\frac{k_i j z_i}{n}\Big) \Big) \tag{62}$$

$$= -1 + \frac{1}{n} \sum_{j=0}^{n-1} \prod_{i=1}^{d} \Big( \sum_{k_i \in \{1, \cdots, n\}} \big( \sum_{q_i \equiv k_i \bmod n} \gamma_\alpha(q_i) \big) \exp\Big(2\pi\mathbf{i}\frac{k_i j z_i}{n}\Big) \Big) \tag{63}$$

Now, we check the term $\sum_{k_i \in \{1, \cdots, n\}} \big( \sum_{q_i \equiv k_i \bmod n} \gamma_\alpha(q_i) \big)$. From the definition of the function $\gamma_\alpha(\cdot)$, for $\forall k_i \in \{1, \cdots, n\}$, we have that

$$\chi(k_i) = \sum_{q_i \equiv k_i \bmod n} \gamma_\alpha(q_i) = \begin{cases} 1 + 2\sum_{m=1}^{\infty} \frac{1}{(mn)^\alpha} & \text{if } k_i = n \\ \sum_{m=0}^{\infty} \frac{1}{(k_i+mn)^\alpha} + \sum_{m=0}^{\infty} \frac{1}{(n-k_i+mn)^\alpha} & \text{otherwise} \end{cases} \tag{64}$$

Note that series $\sum_{m=1}^{\infty} \frac{1}{(mn)^\alpha}$, $\sum_{m=0}^{\infty} \frac{1}{(k_i+mn)^\alpha}$ and $\sum_{m=0}^{\infty} \frac{1}{(n-k_i+mn)^\alpha}$ can be rewritten as

$$\sum_{m=1}^{\infty} \frac{1}{(mn)^\alpha} = \frac{1}{n^\alpha} \sum_{m=1}^{\infty} \frac{1}{m^\alpha} = \frac{1}{n^\alpha}\zeta(\alpha, 1) \tag{65}$$

$$\sum_{m=0}^{\infty} \frac{1}{(k_i + mn)^\alpha} = \frac{1}{n^\alpha} \sum_{m=0}^{\infty} \frac{1}{(\frac{k_i}{n} + m)^\alpha} = \frac{1}{n^\alpha}\zeta(\alpha, \frac{k_i}{n}) \tag{66}$$

$$\sum_{m=0}^{\infty} \frac{1}{(n - k_i + mn)^\alpha} = \frac{1}{n^\alpha} \sum_{m=0}^{\infty} \frac{1}{(\frac{n-k_i}{n} + m)^\alpha} = \frac{1}{n^\alpha}\zeta(\alpha, \frac{n-k_i}{n}) \tag{67}$$

where $\zeta(\cdot, \cdot)$ denotes the Hurwitz zeta function.

Plug them into Eq.(64), we know that

$$\chi(k_i) = \sum_{q_i \equiv k_i \bmod n} \gamma_\alpha(q_i) = \begin{cases} 1 + \frac{2}{n^\alpha}\zeta(\alpha, 1) & \text{if } k_i = n \\ \frac{1}{n^\alpha}\big(\zeta(\alpha, \frac{k_i}{n}) + \zeta(\alpha, \frac{n-k_i}{n})\big) & \text{otherwise} \end{cases} \tag{68}$$

Plug Eq.(68) into Eq.(63), we have that

$$e^2(\mathcal{H}_k; \mathcal{P}) = -1 + \frac{1}{n} \sum_{j=0}^{n-1} \prod_{i=1}^{d} \Big( \sum_{k_i \in \{1, \cdots, n\}} \chi(k_i) \exp\Big(2\pi\mathbf{i}\frac{k_i j z_i}{n}\Big) \Big) \tag{69}$$

$\square$

**Lemma 3.** *Let $n$ be a prime number. Let $\boldsymbol{\gamma} = [\gamma_1, \cdots, \gamma_n]^\top$ be a vector with $\gamma_k = \chi(k)$ for $k \in \{1, \cdots, n\}$, where $\chi(\cdot)$ is defined in Lemma 2. The square worst-case integral approximation error of rank-1 lattice $\mathcal{P}$ constructed by generating vector $\boldsymbol{z} = [z_1, \cdots, z_d]$ can be rewritten in a matrix form as Eq.(70)*

$$e^2(\mathcal{H}_k; \mathcal{P}) = \frac{1}{n} \mathbf{1}^\top \left( \boldsymbol{h}^0 \odot \cdots \odot \boldsymbol{h}^{d-1} - \mathbf{1} \right) \tag{70}$$

*where $\odot$ denotes the element-wise product, symbol $\mathbf{1}$ denotes the vector with elements all ones, and $\boldsymbol{h}^i = \boldsymbol{F}^i \boldsymbol{\gamma}$ with $\boldsymbol{F}$ as the discrete Fourier matrix, i.e., $\boldsymbol{F}_{jk} = \exp(2\pi \mathbf{i} \frac{jk}{n})$, and $\boldsymbol{F}^i$ denotes the matrix after permutation of the rows of $\boldsymbol{F}$ such that the $j^{th}$ row of $\boldsymbol{F}^i$ equals to the $\widetilde{j}^{th}$ row of $\boldsymbol{F}$, where $\widetilde{j} = j z_{i+1} \bmod n$.*

*Proof.* Define $\boldsymbol{h}^i$ as Eq.(71)

$$\boldsymbol{h}^i = \boldsymbol{F}^i \boldsymbol{\gamma} \tag{71}$$

where $\boldsymbol{F}$ as the discrete Fourier matrix, i.e., $\boldsymbol{F}_{jk} = \exp(2\pi \mathbf{i} \frac{jk}{n})$, and $\boldsymbol{F}^i$ denotes the matrix after permutation of the rows of $\boldsymbol{F}$ such that the $j^{th}$ row of $\boldsymbol{F}^i$ equals to the $\widetilde{j}^{th}$ row of $\boldsymbol{F}$, where $\widetilde{j} = j z_{i+1} \bmod n$, and $g$ denotes the primitive root modulo $n$.

From Lemma 2, we know that

$$e^2(\mathcal{H}_k; \mathcal{P}) = -1 + \frac{1}{n} \sum_{j=0}^{n-1} \prod_{i=1}^{d} \Big( \sum_{k_i \in \{1, \cdots, n\}} \chi(k_i) \exp\Big( 2\pi \mathbf{i} \frac{k_i j z_i}{n} \Big) \Big) \tag{72}$$

Note that $\boldsymbol{\gamma} = [\gamma_1, \cdots, \gamma_n]^\top$ is a vector with $\gamma_k = \chi(k)$ for $k \in \{1, \cdots, n\}$, it follows that

$$e^2(\mathcal{H}_k; \mathcal{P}) = -1 + \frac{1}{n} \mathbf{1}^\top \left( \boldsymbol{F}^0 \boldsymbol{\gamma} \odot \cdots \odot \boldsymbol{F}^{d-1} \boldsymbol{\gamma} \right) \tag{73}$$

$$= -1 + \frac{1}{n} \mathbf{1}^\top \left( \boldsymbol{h}^0 \odot \cdots \odot \boldsymbol{h}^{d-1} \right) \tag{74}$$

$$= \frac{1}{n} \mathbf{1}^\top \left( \boldsymbol{h}^0 \odot \cdots \odot \boldsymbol{h}^{d-1} - \mathbf{1} \right) \tag{75}$$

$\square$

**Lemma 4.** *Let $n$ be a prime number such that $(2d-1)|(n-1)$. Let $\boldsymbol{\gamma} = [\gamma_1, \cdots, \gamma_n]^\top$ be a vector with $\gamma_k = \chi(k)$ for $k \in \{1, \cdots, n\}$, where $\chi(\cdot)$ is defined in Lemma 2. Given a rank-1 lattice $\mathcal{P}$ constructed by generating vector in Eq.(30), then we have Eq.(76)*

$$\mathbf{1}^\top (\boldsymbol{h}^0 \odot \cdots \odot \boldsymbol{h}^{d-1} - \mathbf{1}) = \mathbf{1}^\top (\boldsymbol{h}^d \odot \cdots \odot \boldsymbol{h}^{2d-2} - \mathbf{1}) + \langle \boldsymbol{h}^d \odot \cdots \odot \boldsymbol{h}^{2d-2} - \mathbf{1}, \boldsymbol{h}^0 - \mathbf{1} \rangle$$
$$+ \mathbf{1}^\top (\boldsymbol{h}^0 - \mathbf{1}) \tag{76}$$

*where $\odot$ denotes the element-wise product, symbol $\mathbf{1}$ denotes the vector with elements all ones, and $\boldsymbol{h}^i = \boldsymbol{F}^i \boldsymbol{\gamma}$ with $\boldsymbol{F}$ as the discrete Fourier matrix, i.e., $\boldsymbol{F}_{jk} = \exp(2\pi \mathbf{i} \frac{jk}{n})$, and $\boldsymbol{F}^i$ denotes the matrix after permutation of the rows of $\boldsymbol{F}$ such that the $j^{th}$ row of $\boldsymbol{F}^i$ equals to the $\widetilde{j}^{th}$ row of $\boldsymbol{F}$, where $\widetilde{j} = j g^{\frac{i(n-1)}{2d-1}} \bmod n$, and $g$ denotes the primitive root modulo $n$.*

*Proof.* Note that $\boldsymbol{h}^i = \boldsymbol{F}^i \boldsymbol{\gamma}$ is a permutation of $\boldsymbol{h}^0$. From the definition of permutation $\boldsymbol{F}^i$, we know that the $j^{th}$ row of $\boldsymbol{F}^i$ equals to the $\widetilde{j}^{th}$ row of $\boldsymbol{F}$ with $\widetilde{j} = j g^{\frac{i(n-1)}{2d-1}} \bmod n$. Note that $(2d-1)|(n-1)$ and $n$ is a prime number, we know $\{1, g^{\frac{1(n-1)}{2d-1}}, \cdots, g^{\frac{(2d-2)(n-1)}{2d-1}}\}$ modulo $n$ forms a subgroup of $\{1, \cdots, n-1\}$ modulo $n$. Thus, we know $\{\boldsymbol{h}^0, \boldsymbol{h}^1, \cdots, \boldsymbol{h}^{2d-2}\}$ forms a group, and $\boldsymbol{h}^0 = \boldsymbol{h}^{2d-1}$. Furthermore, we know that $\boldsymbol{h}^k$ is a permutation of $\boldsymbol{h}^i$ such that $j^{th}$ row of $\boldsymbol{F}^k$ equals to the $\bar{j}^{th}$ row of $\boldsymbol{F}^i$ with $\bar{j} = j g^{\frac{(k-i)(n-1)}{2d-1}} \bmod n$. Thus, we know that

$$\mathbf{1}^\top (\boldsymbol{h}^0 \odot \cdots \odot \boldsymbol{h}^{d-1}) = \mathbf{1}^\top (\boldsymbol{h}^d \odot \cdots \odot \boldsymbol{h}^{2d-1}) \tag{77}$$

Note that $h^0 = h^{2d-1}$. It follows that

$$1^\top(h^0 \odot \cdots \odot h^{d-1} - 1) = 1^\top(h^d \odot \cdots \odot h^{2d-1} - 1) \tag{78}$$

$$= 1^\top(h^d \odot \cdots \odot h^{2d-2} \odot h^0 - 1) \tag{79}$$

In addition, we have that

$$\langle h^d \odot \cdots \odot h^{2d-2} - 1, h^0 - 1 \rangle \tag{80}$$

$$= \langle h^d \odot \cdots \odot h^{2d-2}, h^0 \rangle - 1^\top(h^d \odot \cdots \odot h^{2d-2}) - 1^\top h^0 + 1^\top 1 \tag{81}$$

$$= 1^\top(h^d \odot \cdots \odot h^{2d-2} \odot h^0) - 1^\top(h^d \odot \cdots \odot h^{2d-2}) - 1^\top h^0 + 1^\top 1 \tag{82}$$

$$= 1^\top(h^d \odot \cdots \odot h^{2d-2} \odot h^0) - 1^\top 1 - 1^\top(h^d \odot \cdots \odot h^{2d-2}) + 1^\top 1 - 1^\top h^0 + 1^\top 1 \tag{83}$$

$$= 1^\top(h^d \odot \cdots \odot h^{2d-2} \odot h^0 - 1) - 1^\top(h^d \odot \cdots \odot h^{2d-2} - 1) - 1^\top(h^0 - 1) \tag{84}$$

It follows that

$$1^\top(h^d \odot \cdots \odot h^{2d-2} \odot h^0 - 1) = \langle h^d \odot \cdots \odot h^{2d-2} - 1, h^0 - 1 \rangle + 1^\top(h^d \odot \cdots \odot h^{2d-2} - 1)$$
$$+ 1^\top(h^0 - 1) \tag{85}$$

Plug Eq.(85) into Eq.(79), we know that

$$1^\top(h^0 \odot \cdots \odot h^{d-1} - 1) = \langle h^d \odot \cdots \odot h^{2d-2} - 1, h^0 - 1 \rangle + 1^\top(h^d \odot \cdots \odot h^{2d-2} - 1)$$
$$+ 1^\top(h^0 - 1) \tag{86}$$

$\square$

**Lemma 5.** *Let $n$ be a prime number such that $(2d - 1)|(n - 1)$. Let $\gamma = [\gamma_1, \cdots, \gamma_n]^\top$ be a vector with $\gamma_k = \chi(k)$ for $k \in \{1, \cdots, n\}$, where $\chi(\cdot)$ is defined in Lemma 2. Given a rank-1 lattice $\mathcal{P}$ constructed by generating vector in Eq.(30), then we have Eq.(87)*

$$1^\top(h^0 \odot \cdots \odot h^{2d-2} - 1) = 1^\top(h^0 \odot \cdots \odot h^{d-1} - 1) + 1^\top(h^d \odot \cdots \odot h^{2d-2} - 1)$$
$$+ \langle h^0 \odot \cdots \odot h^{d-1} - 1, h^d \odot \cdots \odot h^{2d-2} - 1 \rangle \tag{87}$$

*where $\odot$ denotes the element-wise product, symbol $1$ denotes the vector with elements all ones, and $h^i = F^i \gamma$ with $F$ as the discrete Fourier matrix, i.e., $F_{jk} = \exp(2\pi \mathbf{i} \frac{jk}{n})$, and $F^i$ denotes the matrix after permutation of the rows of $F$ such that the $j^{th}$ row of $F^i$ equals to the $\widetilde{j}^{th}$ row of $F$, where $\widetilde{j} = jg^{\frac{i(n-1)}{2d-1}} \mod n$, and $g$ denotes the primitive root modulo $n$.*

*Proof.* Similar to the proof of Lemma 4, we have that

$$\langle h^0 \odot \cdots \odot h^{d-1} - 1, h^d \odot \cdots \odot h^{2d-2} - 1 \rangle \tag{88}$$

$$= \langle h^0 \odot \cdots \odot h^{d-1}, h^d \odot \cdots \odot h^{2d-2} \rangle - 1^\top(h^0 \odot \cdots \odot h^{d-1}) - 1^\top(h^d \odot \cdots \odot h^{2d-2}) + 1^\top 1 \tag{89}$$

$$= 1^\top(h^0 \odot \cdots \odot h^{2d-2}) - 1^\top(h^0 \odot \cdots \odot h^{d-1}) - 1^\top(h^d \odot \cdots \odot h^{2d-2}) + 1^\top 1 \tag{90}$$

$$= 1^\top(h^0 \odot \cdots \odot h^{2d-2}) - 1^\top 1 - 1^\top(h^0 \odot \cdots \odot h^{d-1}) + 1^\top 1 - 1^\top(h^d \odot \cdots \odot h^{2d-2}) + 1^\top 1 \tag{91}$$

$$= 1^\top(h^0 \odot \cdots \odot h^{2d-2} - 1) - 1^\top(h^0 \odot \cdots \odot h^{d-1} - 1) - 1^\top(h^d \odot \cdots \odot h^{2d-2} - 1) \tag{92}$$

It follows that

$$1^\top(h^0 \odot \cdots \odot h^{2d-2} - 1) = 1^\top(h^0 \odot \cdots \odot h^{d-1} - 1) + 1^\top(h^d \odot \cdots \odot h^{2d-2} - 1)$$
$$+ \langle h^0 \odot \cdots \odot h^{d-1} - 1, h^d \odot \cdots \odot h^{2d-2} - 1 \rangle \tag{93}$$

$\square$

**Lemma 6.** *Let $n$ be a prime number such that $(2d-1)|(n-1)$. Let $\boldsymbol{\gamma} = [\gamma_1, \cdots, \gamma_n]^\top$ be a vector with $\gamma_k = \chi(k)$ for $k \in \{1, \cdots, n\}$, where $\chi(\cdot)$ is defined in Lemma 2. Given a rank-1 lattice $\mathcal{P}$ constructed by generating vector in Eq.(30), then we have Eq.(94)*

$$\mathbf{1}^\top(\boldsymbol{h}^0 \odot \cdots \odot \boldsymbol{h}^{2d-2} - \mathbf{1}) = \mathbf{1}^\top \big((\boldsymbol{h}^1 \odot \cdots \odot \boldsymbol{h}^{d-1} - \mathbf{1}) \odot \boldsymbol{h}^0 \odot (\boldsymbol{h}^{-(d-1)} \odot \cdots \odot \boldsymbol{h}^{-1} - \mathbf{1})\big)$$
$$+ 2\mathbf{1}^\top(\boldsymbol{h}^0 \odot \cdots \odot \boldsymbol{h}^{d-1} - \mathbf{1}) - \mathbf{1}^\top(\boldsymbol{h}^0 - \mathbf{1}) \tag{94}$$

*where $\odot$ denotes the element-wise product, symbol $\mathbf{1}$ denotes the vector with elements all ones, and $\boldsymbol{h}^i = \boldsymbol{F}^i \boldsymbol{\gamma}$ with $\boldsymbol{F}$ as the discrete Fourier matrix, i.e., $\boldsymbol{F}_{jk} = \exp(2\pi \mathbf{i} \frac{jk}{n})$, and $\boldsymbol{F}^i$ denotes the matrix after permutation of the rows of $\boldsymbol{F}$ such that the $j^{th}$ row of $\boldsymbol{F}^i$ equals to the $\widetilde{j}^{th}$ row of $\boldsymbol{F}$, where $\widetilde{j} = jg^{\frac{i(n-1)}{2d-1}} \bmod n$, and $g$ denotes the primitive root modulo $n$.*

*Proof.* Plug Eq.(76) in Lemma 4 into Eq.(87) in Lemma 5, we know that

$$\mathbf{1}^\top(\boldsymbol{h}^0 \odot \cdots \odot \boldsymbol{h}^{2d-2} - \mathbf{1}) = 2\mathbf{1}^\top(\boldsymbol{h}^0 \odot \cdots \odot \boldsymbol{h}^{d-1} - \mathbf{1}) - \mathbf{1}^\top(\boldsymbol{h}^0 - \mathbf{1}) - \langle \boldsymbol{h}^d \odot \cdots \odot \boldsymbol{h}^{2d-2} - \mathbf{1}, \boldsymbol{h}^0 - \mathbf{1}\rangle$$
$$+ \langle \boldsymbol{h}^0 \odot \cdots \odot \boldsymbol{h}^{d-1} - \mathbf{1}, \boldsymbol{h}^d \odot \cdots \odot \boldsymbol{h}^{2d-2} - \mathbf{1}\rangle \tag{95}$$

Now we check the last two terms in Eq.(95). Note that

$$\langle \boldsymbol{h}^0 \odot \cdots \odot \boldsymbol{h}^{d-1} - \mathbf{1}, \boldsymbol{h}^d \odot \cdots \odot \boldsymbol{h}^{2d-2} - \mathbf{1}\rangle - \langle \boldsymbol{h}^d \odot \cdots \odot \boldsymbol{h}^{2d-2} - \mathbf{1}, \boldsymbol{h}^0 - \mathbf{1}\rangle \tag{96}$$
$$= \langle \boldsymbol{h}^0 \odot \cdots \odot \boldsymbol{h}^{d-1} - \mathbf{1} - (\boldsymbol{h}^0 - \mathbf{1}), \boldsymbol{h}^d \odot \cdots \odot \boldsymbol{h}^{2d-2} - \mathbf{1}\rangle \tag{97}$$
$$= \langle \boldsymbol{h}^0 \odot \cdots \odot \boldsymbol{h}^{d-1} - \boldsymbol{h}^0, \boldsymbol{h}^d \odot \cdots \odot \boldsymbol{h}^{2d-2} - \mathbf{1}\rangle \tag{98}$$
$$= \langle \boldsymbol{h}^0 \odot (\boldsymbol{h}^1 \odot \cdots \odot \boldsymbol{h}^{d-1} - \mathbf{1}), \boldsymbol{h}^d \odot \cdots \odot \boldsymbol{h}^{2d-2} - \mathbf{1}\rangle \tag{99}$$
$$= \mathbf{1}^\top \big((\boldsymbol{h}^1 \odot \cdots \odot \boldsymbol{h}^{d-1} - \mathbf{1}) \odot \boldsymbol{h}^0 \odot (\boldsymbol{h}^d \odot \cdots \odot \boldsymbol{h}^{2d-2} - \mathbf{1})\big) \tag{100}$$

It follows that

$$\mathbf{1}^\top(\boldsymbol{h}^0 \odot \cdots \odot \boldsymbol{h}^{2d-2} - \mathbf{1}) = 2\mathbf{1}^\top(\boldsymbol{h}^0 \odot \cdots \odot \boldsymbol{h}^{d-1} - \mathbf{1}) - \mathbf{1}^\top(\boldsymbol{h}^0 - \mathbf{1})$$
$$+ \mathbf{1}^\top \big((\boldsymbol{h}^1 \odot \cdots \odot \boldsymbol{h}^{d-1} - \mathbf{1}) \odot \boldsymbol{h}^0 \odot (\boldsymbol{h}^d \odot \cdots \odot \boldsymbol{h}^{2d-2} - \mathbf{1})\big) \tag{101}$$

Because $\{\boldsymbol{h}^0, \boldsymbol{h}^1, \cdots, \boldsymbol{h}^{2d-2}\}$ forms a group, and $\boldsymbol{h}^0 = \boldsymbol{h}^{2d-1}$ with a modulo period $2d-1$, we know that

$$\boldsymbol{h}^d \odot \cdots \odot \boldsymbol{h}^{2d-2} = \boldsymbol{h}^{-(d-1)} \odot \cdots \odot \boldsymbol{h}^{-1} \tag{102}$$

Plug Eq.(102) into Eq.(101), we have that

$$\mathbf{1}^\top(\boldsymbol{h}^0 \odot \cdots \odot \boldsymbol{h}^{2d-2} - \mathbf{1}) = 2\mathbf{1}^\top(\boldsymbol{h}^0 \odot \cdots \odot \boldsymbol{h}^{d-1} - \mathbf{1}) - \mathbf{1}^\top(\boldsymbol{h}^0 - \mathbf{1})$$
$$+ \mathbf{1}^\top \big((\boldsymbol{h}^1 \odot \cdots \odot \boldsymbol{h}^{d-1} - \mathbf{1}) \odot \boldsymbol{h}^0 \odot (\boldsymbol{h}^{-d-1} \odot \cdots \odot \boldsymbol{h}^{-1} - \mathbf{1})\big) \tag{103}$$

□

Now, we are ready to prove our main Theorem 1.

*Proof.* From Lemma 3, we know that

$$e^2(\mathcal{H}_k; \mathcal{P}) = \frac{1}{n} \mathbf{1}^\top \left(\boldsymbol{h}^0 \odot \cdots \odot \boldsymbol{h}^{d-1} - \mathbf{1}\right) \tag{104}$$

From Lemma 6, we know that

$$\mathbf{1}^\top(\boldsymbol{h}^0 \odot \cdots \odot \boldsymbol{h}^{d-1} - \mathbf{1})$$
$$= \frac{1}{2} \mathbf{1}^\top \big(\boldsymbol{h}^0 \odot \cdots \odot \boldsymbol{h}^{2d-2} - \mathbf{1} - (\boldsymbol{h}^1 \odot \cdots \odot \boldsymbol{h}^{d-1} - \mathbf{1}) \odot \boldsymbol{h}^0 \odot (\boldsymbol{h}^{-1} \odot \cdots \odot \boldsymbol{h}^{-(d-1)} - \mathbf{1})\big)$$
$$+ \frac{1}{2} \mathbf{1}^\top(\boldsymbol{h}^0 - \mathbf{1}) \tag{105}$$

Plug Eq.(105) into Eq.(104), we have that

$$
\begin{aligned}
& e^2(\mathcal{H}_k; \mathcal{P}) \\
& = \frac{1}{2n} \mathbf{1}^\top \big( \boldsymbol{h}^0 \odot \cdots \odot \boldsymbol{h}^{2d-2} - \mathbf{1} - (\boldsymbol{h}^1 \odot \cdots \odot \boldsymbol{h}^{d-1} - \mathbf{1}) \odot \boldsymbol{h}^0 \odot (\boldsymbol{h}^{-1} \odot \cdots \odot \boldsymbol{h}^{-(d-1)} - \mathbf{1}) \big) \\
& + \frac{1}{2n} \mathbf{1}^\top (\boldsymbol{h}^0 - \mathbf{1})
\end{aligned}
\tag{106}
$$

Note that $\boldsymbol{h}^0 = \boldsymbol{F}\boldsymbol{\gamma}$ and $\boldsymbol{F}$ denotes the discrete Fourier matrix, we have that

$$
\mathbf{1}^\top (\boldsymbol{h}^0 - \mathbf{1}) = \mathbf{1}^\top \boldsymbol{F}\boldsymbol{\gamma} - n \tag{107}
$$
$$
= \boldsymbol{b}^\top \boldsymbol{\gamma} - n \tag{108}
$$

where $\boldsymbol{b} = [0, 0, \cdots, 0, n]^\top$.

Note that the $n^{th}$ element in $\boldsymbol{\gamma}$ is $\gamma_n = 1 + \frac{2}{n^\alpha}\zeta(\alpha, 1)$, where $\zeta(\cdot, \cdot)$ denotes the Hurwitz zeta function. It follows that

$$
\mathbf{1}^\top (\boldsymbol{h}^0 - \mathbf{1}) = \boldsymbol{b}^\top \boldsymbol{\gamma} - n = n + n\frac{2}{n^\alpha}\zeta(\alpha, 1) - n = n\frac{2}{n^\alpha}\zeta(\alpha, 1) \tag{109}
$$

Plug Eq.(109) into Eq.(106), we achieve the result in Theorem 1

$$
\begin{aligned}
& e^2(\mathcal{H}_k; \mathcal{P}) \\
& = \frac{1}{2n} \mathbf{1}^\top \big( \boldsymbol{h}^0 \odot \cdots \odot \boldsymbol{h}^{2d-2} - \mathbf{1} - (\boldsymbol{h}^1 \odot \cdots \odot \boldsymbol{h}^{d-1} - \mathbf{1}) \odot \boldsymbol{h}^0 \odot (\boldsymbol{h}^{-1} \odot \cdots \odot \boldsymbol{h}^{-(d-1)} - \mathbf{1}) \big) \\
& + \frac{1}{n^\alpha}\zeta(\alpha, 1)
\end{aligned}
\tag{110}
$$

$\square$

# B  Benchmark Test Functions

The benchmark test functions employed in section 4.1 are listed in Table 2, which contains multi-mode functions and non-smooth functions that are challenging for optimization.

Table 2: Test functions

| name | function |
| --- | --- |
| Rosenbrock | $\sum_{i=1}^{d-1} \left( 100(x_{i+1} - x_i^2)^2 + (1 - x_i)^2 \right)$ |
| Nesterov | $\frac{1}{4}\lvert x_1 - 1\rvert + \sum_{i=1}^{d-1} \lvert x_{i+1} - 2\lvert x_i\rvert + 1\rvert$ |
| Rastrigin | $10d + \sum_{i=1}^{d} \left( x_i^2 - 10\cos(2\pi x_i) \right)$ |

## C Training Time and Fast Coordinate Search Time

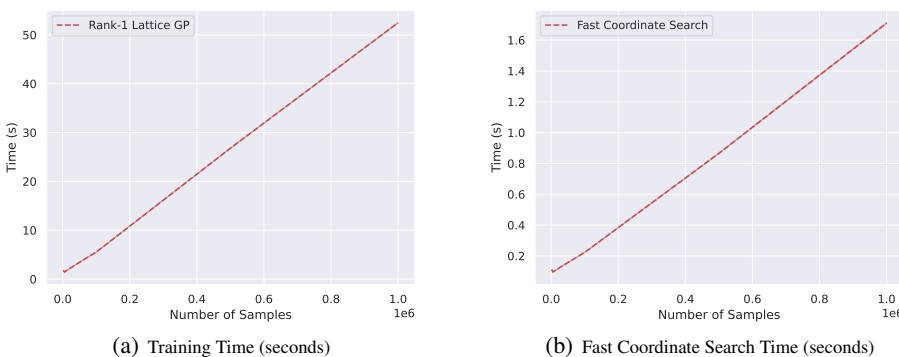

(a) Training Time (seconds)

(b) Fast Coordinate Search Time (seconds)

Figure 5: Training Time and Fast Coordinate Search Time (seconds) v.s. the number of samples

We provide the training time of our rank- 1 lattice GP and the time of our fast coordinate search for targeted sampling in Figure 5(a) and Figure 5(b), respectively. The dimension of the rank-1 lattice data is set to $d = 50$. The number of samples $n$ is set to the parameter in $\{1783, 5347, 10099, 51283, 100189, 501139, 1000099\}$. The number of samples $n$ is a prime number such that $(2d-1)|(n-1)$ to achieve our closed-form rank-1 lattice construction. The number of epochs of training is set to 2000. The number of iterations of fast coordinate search is set to $T = 50$. All the experiments are performed in 50 runs on a single NVIDIA A40 Card.

We report the mean value $\pm$ std in Figure 5. The standard deviation of the time is small. From Figure 5(a), we can see that it takes around 50 seconds for our rank-1 GP training with one million lattice data. Moreover, our fast coordinate search for targeted sampling takes around 1.5 seconds to optimize rank-1 lattice GP posterior prediction conditioned on one million lattice data.

## D Additional Experiments of Black-box Prompt Fine-tuning

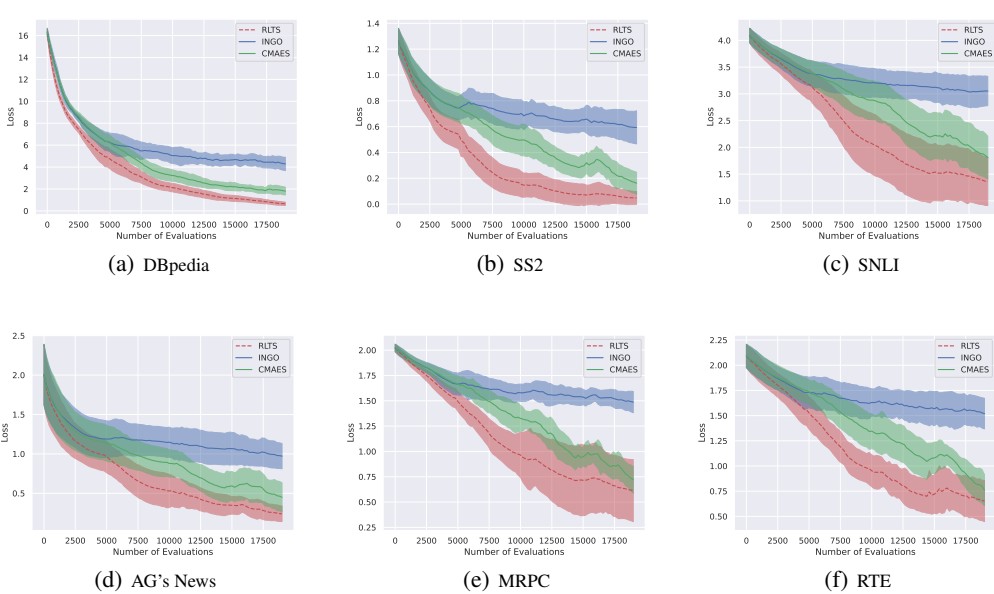

(a) DBpedia

(b) SS2

(c) SNLI

(d) AG's News

(e) MRPC

(f) RTE

Figure 6: Hinge loss v.s. the number of query evaluations on different black-box fine-tuning models.

We provide additional experimental results of black-box prompt fine-tuning for large language models. We employ the deep model in [Sun et al., 2022a] as the backbone. It has 24 layers. For each layer, we set the dimension of the continuous prompt to 50. Thus, the total dimension is $24 \times 50$. We employ the hinge loss of training data as the black-box objective.

In all the experiments, we keep the number of batch samples and the initialization the same for RLTS, INGO and CMAES. We set the number of batch samples to 200. Our RLTS employs 199 rank-1 lattice QMC Gaussian samples and one sample from targeted sampling. INGO employs 199 rank-1 lattice QMC Gaussian samples and one Gaussian sample. CMAES employs 200 Gaussian samples. We initialize the $\boldsymbol{\mu} = \mathbf{0}$ for all the methods. For INGO and RLTS, we set the step-size parameter $\beta = 0.2$ in all experiments. For RLTS, we set the parameter $\eta = 1$ in all experiments. All the experiments are performed in five independent runs with seeds in $\{1, 2, 3, 4, 5\}$.

The experimental results of mean objective $\pm$ std v.s. the number of queries are shown in Figure 6. From Figure 6, we can observe that RLTS decreases the loss consistently faster than INGO and CMAES on all benchmark datasets. More importantly, RLTS decreases the loss significantly faster than INGO. Note that RLTS employs INGO as the backbone algorithm, which shows that RLTS improves the query efficiency of INGO.

