# OpenReview forum: "Fast Rank-1 Lattice Targeted Sampling for Black-box Optimization"
_NeurIPS.cc/2023/Conference — NeurIPS 2023 poster_

### Official Review · Reviewer_RzD5 · 2023-07-02

**Soundness:** 2 fair
**Presentation:** 2 fair
**Contribution:** 2 fair
**Rating:** 6
**Confidence:** 3

**Summary:**

The paper presents the Rank-1 Lattice Targeted Sampling (RLTS) technique as a solution to the challenge of efficiently scaling up black-box optimization in high-dimensional problems. The primary objective of this technique is to enhance the query efficiency while ensuring fast Gaussian process (GP) training and targeted sampling. Additionally, the authors derive a closed-form rank-1 lattice that facilitates rapid lattice sampling. By leveraging these contributions, the paper addresses the need for efficient and scalable optimization methods in high-dimensional settings.

**Strengths:**

The authors of this paper address a crucial issue regarding the performance of Quasi-Monte Carlo methods in high-dimensional problems. They emphasize the challenges associated with effectively applying these methods in such scenarios, as high-dimensional optimization poses significant difficulties and requires a larger number of samples for accurate results. To overcome these challenges, the authors propose an alternative method that aims to enhance efficiency and mitigate the slow convergence of traditional Monte Carlo methods in high dimensions. Their proposed approach offers a solution to improve the performance of Quasi-Monte Carlo methods in high-dimensional optimization tasks.

**Weaknesses:**

- While the paper explores important concepts, it could benefit from improved cohesion and clarity in terms of presenting technical details and providing necessary context.
   - The paper lacks in-depth descriptions of certain important details. For instance, a more comprehensive discussion on lattice methods and low-discrepancy sequences like Sobol sequence, would provide a solid introduction to the topic and enable readers to grasp the advantages of the rank-$1$ lattice approach more clearly. While the advantages of rank-$1$ lattices might be apparent to the authors, it is crucial to explicitly explain these benefits to the readers, ensuring that the significance of this method is clearly conveyed.
   - Another aspect that requires clarification is the initial set of points, denoted as $x \in \mathcal{X}$, and the input box boundaries of the problem. The paper does not explicitly discuss how these initial points should be obtained or provide information regarding the problem's boundaries. Moreover, the process of obtaining the observed values $y$ is only mentioned in Algorithm 2, leaving questions regarding the data acquisition process in the main text.
- The paper acknowledges the challenges faced by Bayesian optimization (BO) when applied to high-dimensional problems. However, the paper lacks a thorough comparison between the proposed RLTS technique and existing BO methods designed for addressing such problems, such as 1) trust regions, 2) structured kernels, 3) subset of active dimensions, or random projections [1, 2, 3]. The paper have not clearly articulated the unique features or advantages that make the RLTS approach competitive with established BO methods. Without explicit confirmation or evidence, it is unclear whether the proposed method performs better, equally, or worse than existing BO techniques in high-dimensional scenarios. Consequently, the comparative performance and potential advantages of the proposed RLTS technique in relation to BO methods remain uncertain.
- The paper does not provide sufficient details regarding their experimental setup, which raises questions about key aspects such as the selection of parameter $T$, the number of data points $n$ involved in the problem, and the specifics of Gaussian process (GP) training. In order to substantiate their claim of accelerated methods, it would be beneficial for the authors to include plots that demonstrate the performance with respect to both time and iterations.

[1] Binois, Mickael and Wycoff, Nathan, "A survey on high-dimensional Gaussian process modeling with application to Bayesian optimization", 2022, https://arxiv.org/abs/2111.05040 \
[2] Diouane, Youssef and Picheny, Victor and Riche, Rodolophe Le and Perrotolo, Alexandre Scotto Di, "TREGO: a trust-region framework for efficient global optimization", 2023, https://arxiv.org/abs/2101.06808 \
[3] Eriksson, David and Pearce, Michael and Gardner, Jacob and Turner, Ryan D and Poloczek, Matthias, "Scalable global optimization via local Bayesian optimization", 2019, https://arxiv.org/abs/1910.01739

**Questions:**

- Although the paper mentions the complexity of the RLTS technique as $n \log(n)$, it does not provide a comprehensive analysis of the overall computational costs. For instance, the targeted search (Algorithm 1) exhibits a time complexity of $O(T d n \log(n))$, and Algorithm 2 involves computationally expensive cubic operations, such as computing the Cholesky decomposition or the square root of the covariance matrix denoted as $\Sigma^{1/2}$, for calculating the weights $w$ (as shown in Equation 25) and updating $\Sigma$. These operations contribute to an overall time complexity of $O(n^3)$ of a sigle iteration of Algorithm 2. Given these complexities, the question arises regarding the specific contribution of the RLTS speed-up to the efficiency of the overall algorithm.
- Could the authors please provide a more detailed explanation of the derivation process for equations 18 and 19? The absence of a thorough explanation and references raises concerns regarding the steps involved in combining the Fourier transform of $y$ with the Fourier transform of the kernel $k$. Alternatively, if a reference exists that explains the derivation, kindly provide the appropriate citation.
- It would be beneficial for the authors to conduct a comprehensive comparison study to evaluate and elucidate the strengths and limitations of their method relative to existing BO techniques in high-dimensional settings. It is also improtant to demonstrate that the RLTS addresses the high-dimensionality problem.
- I would appreciate clarification from the authors regarding the stationarity requirement in the paper. While they assert that the kernel $k_\Delta$ should be stationary (shift invariant or translation invariant), it seems that the stationarity property should be satisfied by the function $\phi$ instead. Since the kernel $k_\Delta$ depends on the distance between data points, which is determined by the output of the $\phi$ function.

---

> ### Author Rebuttal · Authors · 2023-08-05
>
> We sincerely appreciate the reviewer's constructive advice and detailed comments. However, we respectfully disagree that some of the comments may stem from a misunderstanding.  We will kindly clarify the reviewer's misunderstanding first. Afterward, we will provide thorough and detailed responses to the reviewer's questions.
>
> ### Addressing Potential Misunderstandings
>
> M1. "Another aspect that requires clarification is the initial set of points, and the input box boundaries of the problem. The paper does not explicitly discuss how these initial points should be obtained or provide information regarding the problem's boundaries. ...."
>
> C1. Our RLTS replaces the Gaussian sampling in stochastic optimization (or Evolutionary Strategy)  via the inverse cumulative density function. The stochastic optimization (or ES) methods can deal with unconstrained problems. They do not require additional box boundary information like Bayesian Optimization. In our experiments, all the problems are unconstrained optimization problems. The initial parameter $\boldsymbol{\mu}$ of RLTS, INGO, and CMAES is set to $\boldsymbol{\mu}=\boldsymbol{0}$ in all the experiments. The initial covariance matrix $\boldsymbol{\Sigma}$ is set to $\boldsymbol{\Sigma=I}$. The detailed experimental setup can be found in the Appendix of our submission.
>
> M2. "Algorithm 2 involves computationally expensive cubic operations such as computing the Cholesky decomposition or the square root of the covariance matrix denoted as $\Sigma^{1/2}$, for calculating the weights $w$ (as shown in Equation 25) and updating $\Sigma$. These operations contribute to an overall time complexity of $O(n^3)$ of a sigle iteration of Algorithm 2.  ..."
>
> C2. $w={(K_\theta+\sigma^2 I)}^{-1}$ in Eq.(25) can be computed via FFT as $w=\text{ifft}(\text{fft}(y)/\text{fft}(k_\Delta))$ shown in Eq.(22), time complexity is $O(n\log n)$ instead of $O(n^3)$. In addition, the size of $\Sigma^{1/2}$ is $d\times d$. The time complexity to compute $\Sigma^{1/2}$ is $O(d^3)$ instead of $O(n^3)$. In our experiments, we employ the diagonal matrix $\Sigma$ instead of the full matrix with an $O(d)$ time complexity.   Overall, the RLTS does not involve the time-consuming $O(n^3)$ operation, which is crucial to be a plug-in of ES-type algorithms for improving query efficiency. Otherwise,  the algorithm will be too computationally expensive.
>
> ### Responses to Reviewer's Questions
>
> Q1 “... Given these complexities, the question arises regarding the specific contribution of the RLTS speed-up to the efficiency of the overall algorithm.”
>
> A1. As clarified in our response C2, RLTS does not involve the $O(n^3)$ operation. Additionally, we have observed that Algorithm 1 demonstrates rapid convergence with a small T, and we maintain T=5 consistently across all experiments. Empirically, we showcase in Table 1 of the rebuttal file that RLTS achieves a remarkable 300 times speedup compared to TuRBO [3], as suggested by the reviewer.
> It is essential to remark that our RLTS effectively reduces the time complexity of the predominant operation from $O(n^3)$ to $O(n \log n)$, rendering plug-in Evolution Strategies (ES)-type algorithms computationally viable. Without our speedup technique, the overall algorithm would be prohibitively expensive. Furthermore, our RLTS significantly improves the query efficiency compared with the baselines on various benchmark test functions and extensive real-world fine-tuning tasks.
>
> Q2. “Could the authors please provide a more detailed explanation of the derivation process for Eq. 18 and 19?”
>
> A2.  The derivation is based on the circulant matrix's diagonalization and eigenvalue formula.  For a $n\times n$ sized circulant matrix $C$ generated by a vector $c$, we can diagonalize $C$ as  $C= \frac{1}{n}F^* \Lambda F$, where  the $j^{th}$ row and $k^{th}$column element of $F$ is $F_{jk}=e^{-2 \pi jk \boldsymbol{i}/n}$. The matrix $\Lambda$ is a diagonal matrix with diagonal elements as the eigenvalues. $\Lambda$ can be achieved by $\Lambda=diag(Fc)$. The matrix-vector product $Fc$ can be computed by FFT, and the matrix-vector product $\frac{1}{n}F^*b$ for a vector $b$ can be computed by the inverse FFT.  More details about the circulant matrix can be found in the review book [4].
>
> Note that the kernel Gram matrix $K_\theta + \sigma^2 I$ is a symmetric circulant matrix generated by vector $k_\Delta$,  we know $(K_\theta + \sigma^2 I)^{-1}y= \frac{1}{n}F^*\Lambda_k^{-1}Fy$, where $\Lambda_k=diag(F k_\Delta)$.  Then, we can easily achieve Eq.(18) and Eq.(19).    Also, we clean a typo in Eq.(19), the RHS of Eq.(19) should be  $\boldsymbol{1}^\top \log( \text{fft}(k_\Delta) )$ instead of $\boldsymbol{1}^\top \text{fft}(k_\Delta) $, where $\log(\cdot)$ is an elementwise operation.
>
> [4] Robert M. Gray. Toeplitz and Circulant Matrices: A review.
>
> Q3.  “It would be beneficial for the authors to conduct a comprehensive comparison study to evaluate and elucidate the strengths and limitations of their method relative to existing BO techniques in high-dimensional settings. ...”.
>
> A3. Thanks for the reviewer’s suggestion. We compare our RLTS with the high-dimensional BO method TuRBO [3] suggested by the reviewer.    Experimental details can be found in our overall response. The convergence performance regarding the number of queries is shown in Figure 1.  We can observe that RLTS converges significantly faster than TuRBO on the benchmark test problems.
>
>  The running time comparison is shown in Table 1 in the rebuttal file. RLTS performs remarkably faster than TuRBO, achieving around 300 times speedup regarding running time.  The computational time of Bayesian Optimization usually grows cubically fast as the number of queries increases.  In contrast,  our RLTS reduces the expensive $O(n^3)$ operation to $O(n \log n)$ time complexity,  which enables a fast plug-in of the ES-type algorithms.  Without our speedup technique, the whole algorithm will be too expensive.

---

> > ### Comment · Reviewer_RzD5 · 2023-08-15
> >
> > Thank you for providing clarifications; they have greatly enhanced understanding. Nonetheless, I believe that there is room for further improvement in the communication of your ideas. Specifically, it might be beneficial to emphasise your method from the start, and to walk through the significant speed improvements more explicitly in your discussion. Elaborating on finer points, such as explicitly demonstrating the costs of operations involved and illustrating the connections within the algorithm and its derivations, could significantly increase clarity.
> >
> > Moreover, the results achieved with the TuRBO benchmark look really good. However, TuRBO's underwhelming performance speed could potentially be addressed by employing Variational Fourier Features GPs [1], or any other GP utilising precomputed orthogonal features. This approach could potentially reduce the computational cost to O(nm). While conducting such type of experiment is not mandatory, it could fortify the confidence in the unique contribution of your paper.
> >
> > As a result of these factors, I am pleased to revise my score upwards.
> >
> > Q: What are the error bars in your plots? The curves appear very smooth, almost as if there are no variations, which is usually not typical for different BO runs. Could you clarify this?
> >
> > [1] Hensman, J., Durrande, N., & Solin, A. (2017). Variational Fourier Features for Gaussian Processes.

---

> > > ### Author Response · Authors · 2023-08-17
> > >
> > > We sincerely appreciate the reviewer for dedicating time and effort to reviewing our responses and contributing to improving this paper.  We will revise the paper accordingly. The technique of  Random Fourier features is an interesting direction for the acceleration Gaussian Process.  However, it may introduce additional approximation errors to the surrogate compared with the exact GP.  In addition, how to perform a fast search of the candidate using a non-convex RFF-based surrogate remains a challenge. Nevertheless, the RFF technique is interesting and may be potential, we will consider it as one of our future works.
> > >
> > > Q1. What are the error bars in your plots? The curves appear very smooth, ... Could you clarify this?
> > >
> > > A1. The std appears around two orders of magnitudes smaller than the mean value of the cumulative minimum. Note that the plot is regarding the cumulative minimum instead of the min in each batch.  The mean and std are computed with respect to the cumulative minimum.   As a result, the std is small, and the curves appear smooth. We list the mean value and std on the Rastrigin10 benchmark problem below.
> > >
> > > $\textbf{RLTS}$
> > >
> > > | Number of Queries  |    Mean  |  Std |
> > > |  --- |  ---|  ----|
> > > | 2000   |  217862.21    |   1480.36 |
> > > | 4000   |  173428.02   |    2103.94 |
> > > | 6000   |  136313.25   |    1697.49 |
> > > | 8000    | 107996.90   |    1233.35 |
> > > | 10000  |  89482.48    |    1268.98 |
> > > | 12000   |  73045.16   |     586.25 |
> > > | 14000   |  59629.90    |    339.09 |
> > > | 16000   | 50617.44     |   1033.59 |
> > > | 18000   |  41655.20    |    1034.57 |
> > > | 20000   |  35207.45     |   792.14 |
> > > | 22000   |  30141.94     |   583.06 |
> > > | 24000   |  25593.52     |   710.11 |
> > > | 26000   |  22907.04     |    575.04 |
> > > |  28000  |  19083.10    |    665.46 |
> > >  | 30000  |  16948.96     |   485.67 |
> > > | 32000   | 15001.94    |    1069.37 |
> > >  |34000   | 13579.40    |    474.87 |
> > > | 36000   | 12269.29    |    592.90 |
> > > | 38000   |  11290.96   |     550.45 |
> > > | 40000   | 10125.09     |   541.00 |
> > > | 42000   | 9337.53    |     413.24 |
> > > | 44000   |  8718.55    |     348.01 |
> > > | 46000   | 8253.32     |    527.04 |
> > >  | 48000  |  7866.20     |    312.99 |
> > >  | 50000  |  7251.53     |    321.60 |
> > >
> > >
> > > $\textbf{TuRBO}$
> > >
> > > | Number of Queries  |    Mean  |  Std |
> > > |  --- |  ---|  ----|
> > > | 2000   |  205273.02   |    592.74 |
> > > |  4000  |   194977.70   |    868.30 |
> > > | 6000   |  183231.52   |    1387.53 |
> > >  | 8000   |  171598.83   |    1945.68 |
> > > | 10000   | 160583.71    |   1601.47 |
> > >  | 12000   | 150811.57   |    1714.34 |
> > > | 14000   | 142057.02   |    3157.76 |
> > > | 16000  |  133482.36    |   2084.49 |
> > >  | 18000  |  125654.50   |    1385.17 |
> > > |  20000   | 119627.32    |   820.14 |
> > > | 22000   | 113141.73   |    289.69 |
> > > | 24000  |  107433.01    |   797.98 |
> > > | 26000  |  102088.35    |   951.67 |
> > > | 28000  |  98311.84   |     556.94 |
> > >  | 30000  |  94259.84   |      1374.02 |
> > >  | 32000  |  90256.37    |    872.34 |
> > >  | 34000  |  86683.22     |   583.66 |
> > > | 36000   | 82643.07       | 1313.33 |
> > > | 38000  |  80133.17     |   1088.68 |
> > > | 40000   | 77289.02     |   1311.98 |
> > > | 42000  |  75254.05   |     1402.21 |
> > > | 44000   | 73003.95   |     1309.69 |
> > > | 46000   | 70876.92   |     1497.12 |
> > > | 48000   |  68815.14    |    1811.13 |
> > > | 50000  |  66804.85     |   1764.49 |

---

### Official Review · Reviewer_8XeQ · 2023-07-27

**Soundness:** 3 good
**Presentation:** 3 good
**Contribution:** 3 good
**Rating:** 6
**Confidence:** 1

**Summary:**

This paper studies high-dimensional black-box optimization. In order to achieve high efficiency, this paper proposes a Rank-1 Lattice Targeted Sampling (RLTS) technique, using random rank-1 lattice Quasi-Monte Carlo, to perform fast local exact Gaussian processes (GP) training and inference with O(n log n) complexity w.r.t. n batch samples. In addition, this paper developed a fast coordinate searching method with O(n log n) time complexity for fast targeted sampling, improving the query efficiency while scaling up to high dimensional problems. The authors also proposed a closed-form construction to construct rank-1 lattices efficiently.

**Strengths:**

This paper is generally well-structured, and the presentation is clear to the readers. The proposed sampling method for the Black-box optimization problem is novel in this area, which leads to low time complexity and scalability to high dimensional problems. The authors also provide theoretical analysis to show the error pattern of their rank-1 lattice construction. The authors further conduct an empirical evaluation to verify the effectiveness and efficiency of the proposed method on realistic tasks compared to other approaches.

**Weaknesses:**

Since my evaluation of this submission is an educated guess, I would like to modify my rating according to other reviewers' feedback and the authors' response to other reviewers' questions. I have one question regarding this work: In this experiment section, the authors demonstrate the comparison between different approaches via the number of evaluations. Can the authors explain if RLTS still has a faster convergence than other compared methods in terms of physical time?




**Questions:**

Please see the above.

**Limitations:**

The Paper Checklist Guidelines in https://neurips.cc/public/guides/PaperChecklist encourage authors to discuss the limitation, especially in a separate "Limitations" section if possible. But I was unaware of a detailed discussion on the limitations in the paper.

---

> ### Author Rebuttal · Authors · 2023-08-08
>
> We thank the reviewer for dedicating time and effort to review our paper.  Your support is greatly appreciated.
>
> Q1.  Can the authors explain if RLTS still has a faster convergence than other compared methods in terms of physical time?
>
>
> A1. We further compare our RLTS with the high-dimensional BO method,  TuRBO[3]. The experimental convergence results regarding the number of queries and the computational time comparison are shown in Figure 1 and Table 1 in the rebuttal file, respectively.  We can observe that RLTS converges significantly faster than TuRBO.  Remarkably,  RLTS achieves around 300 times speedup compared with TuRBO regarding the computational time.
>
> [3] Eriksson et al. Scalable Global Optimization via Local Bayesian Optimization, NeurIPS 2019.

---

> > ### Comment · Reviewer_8XeQ · 2023-08-18
> >
> > Thank you very much for your response.

---

### Official Review · Reviewer_RbGC · 2023-07-30

**Soundness:** 3 good
**Presentation:** 3 good
**Contribution:** 3 good
**Rating:** 5
**Confidence:** 4

**Summary:**

For the setting of black-box optimization where only function queries are available, this paper propose a rank-1 lattice targeted sampling technique to scale up to high-dimensional problems with good query efficiency. Based on the special structure, this paper use FFT/IFFFT to do fast coordinate search with O(n\log n) time complexity.

**Strengths:**

The authors have a clear background introduction for black-box optimization. Meanwhile, introducing the lattice structure to optimization is new to me. Finally, applying FFT/IFFT with the Lattice structure to get an O(n\log n) time complexity is also impressive.

**Weaknesses:**

The narrative of this paper assumes that the readers have good knowledge of Lattice, but I think that many readers don't have this kind o knowledge. Some concept appears without explanation such as "dual lattice". As a result, it is better to give a more detailed introduction to Lattice, including why this kind of structure can work in the machine-learning context.

Meanwhile, the concept of mod 1 is not clear --- what is the meaning of nonnegative fractional part??? What is the value of a decimal mod 1? Also, what is the correlation between "spaced more evenly " with fast targeted sampling?

Finally, this paper gives a new and interesting structure to reduce complexity, but it doesn't either provide strong convergence results and a strong theoretical verification or provide any meaningful experiments.




,

**Questions:**

See the Weaknesses

---

> ### Author Rebuttal · Authors · 2023-08-07
>
>
> We sincerely appreciate the reviewer's constructive advice and valuable comments.  Detailed responses to the reviewer's questions are given below.
>
> Q1. Some concept appears without explanation such as "dual lattice".
>
> A1.  Thanks for the reviewer’s suggestion. We will revise the paper accordingly.
> The definition of the dual lattice is given in Eq.(38) in the Appendix of our submission.  It is used for the analysis of approximation errors.
>
> Q2. ” Why this kind of structure can work in the machine-learning context?”
>
> A2. Rank-1 Lattice belongs to  Quasi-Monte Carlo(QMC) family. QMC methods are used to approximate integrals or expectations by sampling points in a way that ensures more even coverage of the integration domain compared to traditional random sampling. This improved coverage can lead to more accurate and stable estimates, which in turn can translate into smaller prediction errors when used in the context of Gaussian Processes.
>
> Traditional i.i.d. random sampling may produce regions with very sparse data points, which might miss important features of the function being modeled.  QMC methods can provide a more systematic exploration of such regions, leading to more accurate extrapolation and potentially smaller prediction errors.
>
> In our work, we leverage our rank-1 Lattice to approximate the expectation (gradient approximation), resulting in a more accurate gradient.  Additionally, rank-1 lattices tend to exhibit low GP posterior prediction errors, enhancing the precision of the surrogate model.  This, in turn, empowers our RLTS approach to effectively identify strong candidates.
>
> We would like to emphasize that our subgroup rank-1 lattice capitalizes on constructing a circulant kernel Gram matrix because of its group property. This enables efficient O(n log n) computations in GP training and inference, as well as fast candidate searching. In contrast, low-discrepancy QMC sequences, such as Sobol sequences or Halton sequences, lack these capabilities.
>
> Q3.  What is the meaning of a nonnegative fractional part?
>
> A3.  Given a real number $x$, the nonnegative fractional part of $x$ is  $x- \lfloor x \rfloor $, where $ \lfloor \cdot \rfloor$ denotes the floor function.
>
> Q4.  What is the value of a decimal mod 1?
>
> A4.  Given a real number $x$,  the operation $x$ mod 1 equals $x- \lfloor x \rfloor $.
> We will state it more clearly in our paper.
>
> Q5. What is the correlation between "spaced more evenly " with fast targeted sampling?
>
> A5. A point set that is more evenly spaced helps mitigate very sparse regions, thereby leading to reduced worst-case approximation errors and yielding more accurate and stable estimates. By utilizing our Rank-1 lattice in conjunction with the inverse cumulative density function, we can effectively create evenly spaced points within a Gaussian ball. This helps to reduce the GP posterior prediction error and offers a more precise surrogate for facilitating fast targeted sampling with good candidates.

---

### Official Review · Reviewer_AfkS · 2023-08-02

**Soundness:** 3 good
**Presentation:** 2 fair
**Contribution:** 3 good
**Rating:** 5
**Confidence:** 3

**Summary:**

This paper proposes a novel Rank-1lattice targeted sampling (RLTS) strategy to significantly reduce the GP training & inference time complexity and also enable $O(n\log(n))$ time complexity in order to find a targeting reference point at each iteration. Combined with the evolution strategies, the method proposed in this paper is proven to be efficient in handling high-dimensional problems. The effectiveness of the method is also demonstrated in some benchmark testing functions and a prompt fine-tuning task.

**Strengths:**

(1) The rank-1 lattice sampling strategy reduces the computational/time complexity of GP training/inference significantly in several aspects, which should be interesting to the community

(2) The author also provides a theoretical result to show that the approximation error of this new sampling method is comparable to the integral approximation error.

(3) The effectiveness of the new approach is supported by sufficient experiments.

**Weaknesses:**

(1) Some notation is confusing, for example, $x_i$ in eq(10) and $x_i$ in line 8 of algorithm 2 are different but share the same notation.

(2) For better presentation, the author can consider claiming the assumptions explicitly (for example in sec 3.3 the kernel is required to have a decomposition structure in order to make sure the coordinate searching works)

(3) In GP training, in addition to considering the complexity of computing the value of likelihood, isn't it also important to consider the complexity cost in optimizing the likelihood? Such a result seems to be missing in the paper.



**Questions:**

(1) What is the overall query complexity of Algorithm 2 or how much overall query complexity can RLTS saves compared with INGO? Even if it is a plug-in version of INGO such a result will be helpful for readers to understand the advantage of RLTS over INGO quantively.

(2) How does the author define "time complexity" and "query complexity" in this paper? Are they share the same definition or not? If time complexity refers to the computation cost of solving a concrete optimization sub-problem in the overall algorithm while the query complexity refers to the total number of zeroth-order information that we need, then there is a inconsistency between the motivation and the contribution claimed by the author. (The motivation of using rank-1 sampling seems to target at reducing time complexity while the experiment try to demonstrate that rank-1 sampling can actually reduce query complexity).

**Limitations:**

There no limitation in this paper.

---

> ### Author Rebuttal · Authors · 2023-08-06
>
>
> We sincerely appreciate the reviewer's constructive advice and valuable comments.  Our detailed responses to the reviewer's questions are presented as follows:
>
>
> Q1. “ In GP training, in addition to considering the complexity of computing the value of likelihood, isn't it also important to consider the complexity cost in optimizing the likelihood?”
>
> A1.  In GP training, each gradient descent step typically requires computing the likelihood value (forward pass), resulting in a time complexity of $O(n^3)$.  The backward usually share the same time complexity.  Consequently, the total training complexity is $O(Mn^3)$, where M represents the number of gradient descent steps. This renders standard GP impractical as a plug-in due to its high cost.  In practice, M is often a small constant, and the training complexity is primarily determined by the term related to the number of samples n.
>
> To achieve fast training, it becomes crucial to reduce the complexity of likelihood computation. Our technique successfully reduces this complexity from $O(n^3)$ to $O(n \log n)$, resulting in a more efficient $O(Mn \log n)$ training complexity. This enhancement enables efficient integration with ES-type algorithms. Without our speedup technique, the entire algorithm would be prohibitively expensive.
>
> Q2 “ What is the overall query complexity of Algorithm 2 or how much overall query complexity can RLTS saves compared with INGO? Even if it is a plug-in version of INGO such a result will be helpful for readers to understand the advantage of RLTS over INGO quantively.”
>
> A2. Thanks for the reviewer's suggestion.   In this work, we empirically evaluate RLTS on various benchmark test problems and real-world fine-tuning tasks.   Our experiments demonstrate that RLTS significantly improves query efficiency compared with INGO.   We plan to investigate the convergence rate with respect to the number of queries in our future work.
>
> Q3 “...  then there is a inconsistency between the motivation and the contribution claimed by the author. (The motivation of using rank-1 sampling seems to target at reducing time complexity while the experiment try to demonstrate that rank-1 sampling can actually reduce query complexity).”
>
> A3.  We respectfully disagree with the above comment.  Our main objective is to improve the query efficiency (e.g., the query of the API call in a language model), as stated in the abstract and our contribution statement (third point).  However, directly applying GP as a plug-in poses challenges due to the large training and inference complexity and huge searching cost. The reviewer can refer to Table 1 in our rebuttal file to observe the huge time cost of the GP-based high-dimensional BO baseline TuRBO[3].  Thus, it becomes crucial to develop a method that addresses these issues concurrently. To tackle this, we develop RLTS. Our RLTS reduces the predominant  $O(n^3)$ operation in GP training/inference to $O(n \log n)$, and supports a fast searching algorithm. Empirically, our RLTS significantly improves the query efficiency compared with INGO, CMAES, and TuRBO.  We firmly believe that RLTS presents an important and valuable technique contribution to the field.
>
> [3]  Eriksson et al., "Scalable global optimization via local Bayesian optimization", NeurIPS 2019.

---

### Author Rebuttal · Authors · 2023-08-07

# Overall Response

We sincerely appreciate all reviewers’ constructive advice and valuable comments. We first present an overall response regarding the common concerns raised. Subsequently, we provide detailed responses to each reviewer’s specific questions.

### Clarification

In this work, our main objective is to improve the query efficiency (e.g., the query of the API call in a language model). However, directly applying GP as a plug-in poses great challenges due to the large training and inference complexity, as well as the vast candidate searching cost. The reviewers can refer to Table 1 in our rebuttal file to observe the vast time cost of the GP-based high-dimensional BO baseline TuRBO[3]. Thus, it becomes crucial to develop a method that addresses these issues simultaneously. To tackle this, we develop RLTS. Our RLTS reduces the predominant $O(n^3)$ operation in GP training/inference to $O(n \log n)$, and supports a fast searching algorithm.  Empirically, our RLTS significantly improves the query efficiency compared with INGO, CMAES, and TuRBO.

We would like to emphasize that our subgroup rank-1 lattice capitalizes on constructing a circulant kernel Gram matrix benefit from its group property. This enables efficient O(n log n) computations in GP training/inference, as well as fast candidate searching. In contrast, low-discrepancy QMC sequences, such as Sobol sequences or Halton sequences, lack these capabilities.  We firmly believe that RLTS presents an important and valuable technique contribution to the area. Moreover,  our new closed-form rank-1 lattice may have potential applications in downstream tasks beyond black-box optimization.


### Comparison with High-dimensional BO

We compare our RLTS with the high-dimensional BO method TuRBO [3] suggested by the reviewer. We evaluate RLTS on the three benchmark functions employed in our submission: the Rosenbrock function, the Rastrigin10 function, and the Nesterov function.  The dimension is set to 500. The number of initial points of TuRBO is set to 2000. The batch size of both RLTS and TuRBO is set to 2000. The maximum number of queries is set to 50,000. We employ the default box boundary for TuRBO, i.e.,  $[-5,10]^d$.    The initial parameter $\mu$ of RLTS is set to $\mu=0$, and $\Sigma$ is set to $\Sigma=I$, the same setting in our submission. For TuRBO, we employ the official code provided in the paper [3]. All the methods are performed in three independent runs.

The convergence performance regarding the number of queries is shown in Figure 1 in the rebuttal file. We can observe that RLTS converges faster than TuRBO on the benchmark test problems, demonstrating that RLTS improves query efficiency.

We further report the running time of RLTS and TuRBO on the same machine for evaluation. The results are shown in Table 1 in the rebuttal file. We can observe that RLTS performs significantly faster than TuRBO, achieving around 300 times speedup regarding running time. The computation time of Bayesian Optimization usually grows cubically fast as the number of queries increases. In contrast,  our RLTS reduces the expensive $O(n^3)$ operation to $O(n \log n)$ time complexity,  which enables a fast plug-in of the ES-type algorithms.

In addition, subsets of active dimensions and random projections are techniques for reducing the effective dimension. These techniques are orthogonal to ours. They can be combined to improve performance further.

[3] Eriksson et al.    Scalable Global Optimization via Local Bayesian Optimization, NeurIPS 2019.

---

### Decision · Program_Chairs · 2023-09-21

**Decision:**

Accept (poster)

**Comment:**

This paper considers the block optimization problem. The proposed approach is essentially a zeroth-order approach, using function value to estimate the hessian information for acceleration. To ensure the efficiency, the hessian information is used rank-1 matrix to approximate. The idea looks reasonable and straightforward.

The overall feedback is positive for this paper. The authors solved the concerns in experiment and the computational complexity pointed out in review. We decide to accept this paper. But this paper's theoretical justification can be stronger, e.g.,

- discuss the convergence rate,
- whether converge to the global optimum.

The theoretical supports can refer to the zeroth order optimization literature.